# Singlet fission dynamics modulated by molecular configuration in covalently linked pyrene dimers, *Anti*- and *Syn*-1,2-di(pyrenyl)benzene

Jungkweon Choi[1,2], Siin Kim[1,2], Mina Ahn[3], Jungmin Kim[1,2], Dae Won Cho[1,2], Doyeong Kim[1,2], Seunghwan Eom[1,2], Donghwan Im[1,2], Yujeong Kim [4], Sun Hee Kim[4], Kyung-Ryang Wee[3✉] & Hyotcherl Ihee [1,2✉]

Covalently linked dimers (CLDs) and their structural isomers have attracted much attention as potential materials for improving power conversion efficiencies through singlet fission (SF). Here, we designed and synthesized two covalently *ortho*-linked pyrene (Py) dimers, *anti*- and *syn*-1,2-di(pyrenyl)benzene (*Anti*-DPyB and *Syn*-DPyB, respectively), and investigated the effect of molecular configuration on SF dynamics using steady-state and time-resolved spectroscopies. Both *Anti*-DPyB and *Syn*-DPyB, which have different Py-stacking configurations, form excimers, which then relax to the correlated triplet pair (($T_1T_1$)) state, indicating the occurrence of SF. Unlike previous studies where the excimer formation inhibited an SF process, the ($T_1T_1$)'s of *Anti*-DPyB and *Syn*-DPyB are formed through the excimer state. The dissociation of ($T_1T_1$)'s to $2T_1$ in *Anti*-DPyB is more favorable than in *Syn*-DPyB. Our results showcase that the molecular configuration of a CLD plays an important role in SF dynamics.

[1] Center for Advanced Reaction Dynamics, Institute for Basic Science, Daejeon 34141, Republic of Korea. [2] Department of Chemistry and KI for the BioCentury, Korea Advanced Institute of Science and Technology (KAIST), Daejeon 34141, Republic of Korea. [3] Department of Chemistry and Institute of Natural Science, Daegu University, Gyeongsan 38453, Republic of Korea. [4] Western Seoul Center, Korea Basic Science Institute (KBSI), Seoul 03759, Republic of Korea. ✉email: krwee@daegu.ac.kr; hyotcherl.ihee@kaist.ac.kr

The electron (or charge) carrier dynamics in photoelectric and electrochemical devices are key to determining the performance of devices[1–6]. Chromophore–chromophore interaction as well as the electronic-state coupling of a chromophore can modulate such electron carrier dynamics[7–16]. In this regard, many multi-chromophore systems have been widely used for developing highly efficient photoelectric or electrochemical devices using chromophore–chromophore interaction[7,12–21]. Among the multi-chromophore systems, covalently linked dimers (CLDs) have attracted much attention as potential materials to provide high energy-conversion efficiencies in photovoltaic devices because their excited-state relaxation dynamics, such as the excimer formation, intramolecular charge transfer (ICT), and singlet fission (SF), can be modulated by strategic molecular design[9,15,22–27]. Especially the dynamics of SF, which is a conversion process from one singlet exciton into two triplet excitons, have been actively investigated with various time-resolved spectroscopies to overcome the limit of Shockley–Queisser power conversion efficiency[17,28–32].

The results of extensive studies on SF dynamics and mechanisms showed that they cannot be easily explained by a single unifying mechanism of SF; instead, the intermolecular and intramolecular SF dynamics occur through various types of species such as charge transfer species, excimers, and higher excited vibrational and electronic states, depending on the interactions between chromophores[21,24–26,29,31,33–37]. For example, Zirzlmeier et al. suggested that the SF process occurring in ortho-, meta-, and para-linked pentacene dimers proceed through virtual CT states[29]. Margulies et al. reported that the covalently linked terylene-3,4:11,12-bis(dicarboximide) (TDI) dimer with a stacked structure forms an excimer within <200 fs, whereas the slip-stacked TDI dimer in a nonpolar solvent forms the correlated triplet pair ((T₁T₁)), which is an intermediate in the SF process[30]. They also showed that the slip-stacked TDI dimer forms a CT state

in a few picoseconds in a polar solvent, suggesting that adjusting the CT state energy relative to exciton states via solute–solvent interaction can either promote or inhibit SF[30]. Ni et al. reported that, upon excitation at 250 nm, the cofacial perylene dimer undergoes SF from the upper excited vibrational and electronic states, whereas upon excitation at 450 nm, it fast forms an excimer, which relaxes to the ground state within nanoseconds[38]. Korovina et al. showed that the ortho- and para-bis(ethynyltetracenyl)benzene dimers with a relatively stronger through-bond coupling exhibit more efficient SF dynamics than the meta-bis(ethynyltetracenyl)-benzene dimer[39]. Especially, unlike the ortho-bis(ethynyltetracenyl)benzene dimer, which forms only (T₁T₁), the para-bis(ethynyltetracenyl)benzene dimer shows complete SF dynamics to form free triplets, suggesting that the rotational flexibility between the acenes in the dimers plays an important role in SF dynamics[39]. Shizu et al. demonstrated that the efficient SF dynamics of para-bis(ethynyltetracenyl)benzene dimer is due to large vibronic coupling and the small energy difference between the singlet excited state and the (T₁T₁) state[40]. In contrast, Nakamura et al. showed that in a series of ortho-, meta-, and para-bis(tri isopropyl silylethynyl)-tetracenyl)benzene dimers, the meta-linked tetracene dimer exhibits more efficient SF dynamics than ortho- and para-linked tetracene dimers[15]. They suggested that the large conformational flexibility and weak electronic coupling play a critical role in their SF dynamics[15]. In addition, the experimental and theoretical calculation results of various CLDs showed that compared to ortho- and para-linked dimers, meta-linked dimers exhibit more efficient SF dynamics due to the small binding energy ($E_b$) of (T₁T₁) ($E_b = 2 E|S_0T_1\rangle - E|(T_1T_1)\rangle$)[29,41–43].

Despite numerous experimental and theoretical approaches to determining the SF dynamics of CLDs, a full understanding of the parameters influencing the SF dynamics of CLDs is still lacking. Additionally, studies on the effects of a molecular configuration, which can affect its excited-state relaxation dynamics, may provide clues for the optimized spatial arrangements that ensure the high-energy conversion efficiency of a real device. In this regard, the role of conformational flexibility in the SF dynamics of CLDs has been studied, but still needs further clarification. Accordingly, in-depth studies are needed to understand their excited-state relaxation dynamics, including SF. From this perspective, we designed and synthesized two covalently ortho-linked pyrene (Py) dimers, anti- and syn-1,2-di(pyrenyl)benzene (Anti-DPyB and Syn-DPyB) (see Supplementary Methods and Fig. 1 and Supplementary Figs. S1–S6), which we expected to have different configurations, to elucidate the effect of the molecular configuration for their excited-state relaxation dynamics using steady-state and time-resolved spectroscopies. The data show that both Anti-DPyB and Syn-DPyB form excimers, which rapidly relax to the (T₁T₁) state regardless of solvent polarity, indicating the occurrence of SF dynamics. Notably, the (T₁T₁)s of Anti-DPyB in both n-hexane and acetonitrile dissociate to free triplets as the end product, completing SF, whereas the dissociation of (T₁T₁)'s in Syn-DPyB is less favorable compared with that in Anti-DPyB, indicating that the (T₁T₁) of Syn-DPyB is more bound with respect to the separated triplets than that of Anti-DPyB. Our findings show that the SF dynamics of Anti- and Syn-DPyB differ due to the different molecular configurations of Anti- and Syn-DPyB.

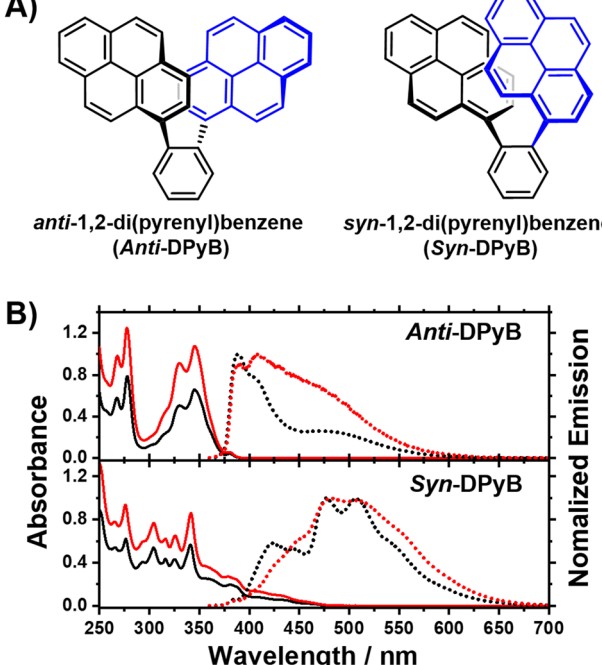

**Fig. 1 Structures and UV–visible absorption (solid line) and emission spectra (dotted line). A** Geometric isomers of anti-1,2-di(phenyl)benzene (Anti-DPyB) and syn-1,2-di(phenyl)benzene (Syn-DPyB). **B** UV–visible absorption spectra of Anti-DPyB and Syn-DPyB in n-hexane (black) and acetonitrile (red). Emission spectra of Anti-DPyB and Syn-DPyB in n-hexane (black) and acetonitrile (red) ($\lambda_{ex} = 345$ nm).

## Results

**Molecular structures**. To characterize the structures of Anti-DPyB and Syn-DPyB, we calculated their minimum energy structures using density functional theory (DFT). The optimized structures are shown in Fig. 1A. Two Py moieties in Anti-DPyB are far from each other, whereas Syn-DPyB shows a partial overlap of two pre-stacked Py moieties. According to the calculations using

B3LYP/6-31G(d,p), the distances between two Py moieties in *Anti*-DPyB and *Syn*-DPyB are 10.326 and 8.481 Å, respectively (see Supplementary Data 2). This result is consistent with the calculation results reported by Jo et al. [44]. The optimized structure of *Anti*-DPyB obtained from our calculation is similar to its crystal structure (Supplementary Fig. S7, Supplementary Tables S1–S6, and Supplementary Data 1).

**Steady-state absorption and emission spectra**. As shown in Fig. 1B, *Anti*-DPyB in *n*-hexane and acetonitrile shows two vibrationally resolved absorption bands at around 250–280 and 300–380 nm. This feature is similar to the absorption spectrum of 1-phenylpyrene (Ph-Py)[9,45]. In contrast, *Syn*-DPyB exhibits a broad and vibrationally resolved absorption band at 250–475 nm. The vibrationally resolved absorption bands observed from *Anti*-DPyB and *Syn*-DPyB indicate that both compounds have highly rigid structures in the ground state. Furthermore, the absorption bands of *Anti*-DPyB and *Syn*-DPyB do not show noticeable dependence on solvent polarity (Fig. 1B).

To investigate the excited-state behaviors of *Anti*-DPyB and *Syn*-DPyB, we measured their emission spectra in *n*-hexane and acetonitrile with excitation at 345 nm, which corresponds to the major absorption peak. *Anti*-DPyB and *Syn*-DPyB in both solvents show dual emission bands (~380 and ~480 nm for *Anti*-DPyB and ~420 and ~480 nm for *Syn*-DPyB), the detailed features of which are different. We checked the possibility that Py molecules are present as impurities in *Anti*-DPyB and *Syn*-DPyB solutions. It is known that the Py molecule shows a strong fluorescence in solutions. Therefore, if Py molecules are present as impurities in *Anti*-DPyB and *Syn*-DPyB solutions, they may contaminate the fluorescence spectra from the *Anti*-DPyB and *Syn*-DPyB samples. To check this possibility, we measured the fluorescence excitation spectra of *Anti*-DPyB and *Syn*-DPyB in acetonitrile at two emission peak positions (380 and 480 nm). As shown in Supplementary Fig. S8, the fluorescence excitation spectra from *Anti*-DPyB and *Syn*-DPyB are significantly different from the absorption spectrum of the Py molecule. This result indicates that Py molecules do not exist as impurities in *Anti*-DPyB and *Syn*-DPyB solutions. The ~380 nm band of *Anti*-DPyB is structured, whereas the broad band centered at ~480 nm is structureless (Fig. 1B). Unlike *Anti*-DPyB, both of the emission bands of *Syn*-DPyB are structured. Based on numerous previous studies on Py and Py derivatives[46,47], the shorter-wavelength (~380 and ~420 nm) emissions from *Anti*-DPyB and *Syn*-DPyB can be assigned to Py monomer moieties, and the longer-wavelength (~480 nm for both) emissions to the excimer formed via the association of excited and unexcited Py's. In addition, the broad structureless bands centered at ~480 nm observed for *Anti*-DPyB are highly similar to the typical excimer bands observed for Py and Py derivatives. In terms of solvent dependence, in *Anti*-DPyB, the relative intensities of the emissions from the monomeric Py moiety and excimer show a significant dependence on solvent polarity, whereas *Syn*-DPyB does not show definite solvent dependency.

To accurately determine the singlet energy ($E_{S1}$) and triplet energy ($E_{T1}$) values of Py, Ph-Py, *Anti*-DPyB, and *Syn*-DPyB, we also measured emission spectra of Py, Ph-Py, *Anti*-DPyB, and *Syn*-DPyB in MTHF containing iodomethane at 77 K. As shown in Supplementary Fig. S9, all four compounds show dual emission bands at around 370–550 and 580–800 nm, corresponding to fluorescence and phosphorescence, respectively. The $E_{S1}$ values of Py, Ph-Py, *Anti*-DPyB, and *Syn*-DPyB are determined to be 3.3, 3.3, 3.3, and 3.1 eV, respectively. From the phosphorescence spectra, the $E_{T1}$ values of Py, Ph-Py, *Anti*-DPyB, and *Syn*-DPyB are determined to be 2.10, 2.03, 2.04, and 1.87 eV, respectively.

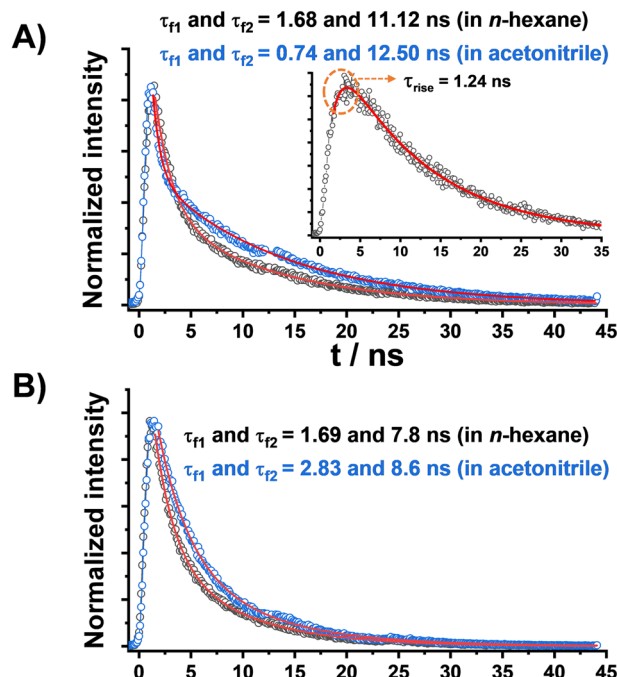

**Fig. 2 Fluorescence decay profiles. A** Fluorescence decay profiles of *Anti*-DPyB in *n*-hexane (black) and acetonitrile (blue), respectively. In **A**, the inset shows the fluorescence decay profile of *Anti*-DPyB measured at the long wavelength (500–600 nm) in *n*-hexane. The fluorescence decay profile of *Anti*-DPyB measured at the long-wavelength (500–600 nm) shows a rise time of 1.24 ns as well as a decay time of 11.1 ns. **B** Fluorescence decay profiles of *Syn*-DPyB in *n*-hexane (black) and acetonitrile (blue), respectively.

**Fluorescence lifetime**. To further elucidate the excited-state relaxation dynamics, we measured the fluorescence decay profiles of *Anti*-DPyB and *Syn*-DPyB in *n*-hexane and acetonitrile. As depicted in Fig. 2, all decay profiles satisfactorily fit with bi-exponential functions. The determined fluorescence lifetimes are summarized in Table 1. The fast ($\tau_{f1}$) and slow ($\tau_{f2}$) time constants are predominantly observed in the shorter- and longer-wavelength emissions, respectively. Since the shorter- and longer-wavelength emissions arise from the Py monomeric unit and excimer, respectively, the fast ($\tau_{f1}$) and slow ($\tau_{f2}$) time constants correspond to the lifetimes of the Py monomeric unit and excimer, respectively. In addition, as depicted in Fig. 2A, the fluorescence decay profile of *Anti*-DPyB measured in the range of 500–600 nm in *n*-hexane shows an additional kinetic component with a rise time of 1.24 ns.

**TA spectra**. To elucidate the excited-state relaxation dynamics, we measured the femtosecond transient absorption (fs-TA) spectra for *Anti*-DPyB and *Syn*-DPyB in *n*-hexane and acetonitrile with 350 nm excitation. The TA spectra of *Anti*-DPyB in *n*-hexane and acetonitrile exhibit broad signals at 400–700 nm, corresponding to the excited-state absorption (ESA) (see Fig. 3 and Supplementary Fig. S10). With time, these broad positive signals transform into structured signals. On the other hand, the TA spectra of *Syn*-DPyB in *n*-hexane and acetonitrile exhibit intense ESA signals at 450–530 nm with a weak absorption tail (550–700 nm) (Fig. 3C and D). To extract further information, we analyzed the TA spectra of *Anti*-DPyB and *Syn*-DPyB using singular value decomposition (SVD) analysis (Supplementary Information). The SVD analysis for the TA spectra of *Anti*-DPyB and *Syn*-DPyB identified four and three significant singular components, respectively (Supplementary Figs. S11 and S12). As shown in Supplementary Fig. S13, the

**Table 1 Emission quantum yields (Φ), emission lifetimes (τ), average emission lifetimes (⟨τ⟩), radiative rate constants ($k_R$), and nonradiative rate constants ($k_{NR}$) of *Anti*-DPyB and *Syn*-DPyB in *n*-hexane and acetonitrile.**

| | | Φ | τ (ns) | | | $k_R{}^a$ (×10$^7$ s$^{-1}$) | $k_{NR}{}^b$ (×10$^7$ s$^{-1}$) |
| --- | --- | --- | --- | --- | --- | --- | --- |
| | | | $\tau_{f1}$ | $\tau_{f2}$ | $\langle\tau\rangle$ | | |
| *Anti*-DPyB | In *n*-hexane | 0.20 | 1.68 ± 0.03 | 11.12 ± 0.11 | 4.23 | 4.73 | 18.9 |
| | In acetonitrile | 0.21 | 0.74 ± 0.02 | 12.50 ± 0.07 | 3.33 | 6.31 | 23.7 |
| *Syn*-DPyB | In *n*-hexane | 0.09 | 1.89 ± 0.04 | 7.76 ± 0.10 | 3.59 | 2.51 | 25.3 |
| | In acetonitrile | 0.14 | 2.93 ± 0.10 | 8.64 ± 0.21 | 4.87 | 2.87 | 17.7 |

$^a k_R = \Phi/\langle\tau\rangle$.
$^b k_{NR} = (1-\Phi)/\langle\tau\rangle$.

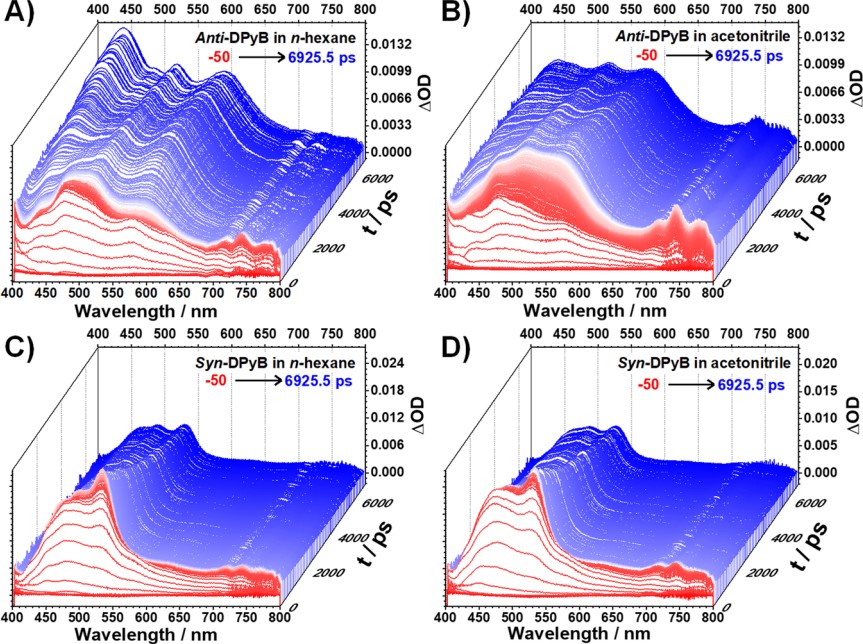

**Fig. 3 Transient absorption spectra. A**, **B** Transient absorption spectra of *Anti*-DPyB in *n*-hexane and acetonitrile, respectively. **C**, **D** Transient absorption spectra of *Syn*-DPyB in *n*-hexane and acetonitrile, respectively.

significant rSVs for *Anti*-DPyB in *n*-hexane and acetonitrile can be expressed by a tetra-exponential function with shared relaxation times (3.6 ± 0.3, 231 ± 19 ps, 1.75 ± 0.12 ns, and >10 ns in *n*-hexane; 2.8 ± 0.1, 24.3 ± 0.5, 495.7 ± 6.5 ps, and >10 ns in acetonitrile). The significant rSVs for *Syn*-DPyB in *n*-hexane and acetonitrile can be expressed by a tri-exponential function with shared relaxation times (2.3 ± 0.8, 9.7 ± 0.5 ps, and 6.4 ± 0.2 ns in *n*-hexane; 2.8, 8.0 ± 0.6 ps, and 4.8 ± 0.2 ns in acetonitrile). The time constants are summarized in Table 2.

To observe the long-lived species, we measured the nanosecond TA spectra of *Anti*-DPyB and *Syn*-DPyB in *n*-hexane and acetonitrile with 355 nm excitation. *Anti*-DPyB in *n*-hexane and acetonitrile shows a weak and broad absorption band around 445 nm at a time delay of a few microseconds (Supplementary Fig. S14), suggesting the presence of a long-lived species. In contrast, *Syn*-DPyB does not exhibit any absorption band in both *n*-hexane, and acetonitrile, indicating that no long-lived species, such as a triplet species, exist at the μs–ms time scale.

**Discussion**
**Intramolecular excimer formation and electronic coupling**. The optimized structures of *Anti*-DPyB and *Syn*-DPyB show that the distance between two Py moieties in *Anti*-DPyB is longer than

that in *Syn*-DPyB, suggesting that the interaction in *Anti*-DPyB between two Py moieties should be weaker than that in *Syn*-DPyB with the partial overlap of two Py moieties. *Anti*-DPyB has an absorption spectrum similar to that of Py or Ph-Py[9,45], indicating that in the ground state, the two Py moieties in *Anti*-DPyB have a monomeric character. In contrast, the pre-stacked structure of *Syn*-DPyB is expected to show the characteristic feature of an excimer. Indeed, *Syn*-DPyB exhibits a single broad and vibrationally resolved absorption band at 250–475 nm owing to the strong π–π interaction between the two Py moieties. The steady-state spectroscopic results confirm this prediction: in the ground state, the interaction between the two Py moieties in *Anti*-DPyB is much weaker than that in *Syn*-DPyB. On the other hand, the cyclic voltammograms for the reduction of *Anti*-DPyB and *Syn*-DPyB in THF show two separate peaks, whereas the cyclic voltammogram of Py exhibits a single peak (Supplementary Fig. S15). From the cyclic voltammograms, the splitting energies ($E_{red2} - E_{red1}$) of *Anti*-DPyB and *Syn*-DPyB are determined to be 0.09 and 0.12 V, respectively (Table 3). Compared to *Syn*-DPyB, the lower splitting energy of *Anti*-DPyB indicates that the electronic coupling between the two Py moieties in *Anti*-DPyB is relatively weaker than that in *Syn*-DPyB. This result is consistent with the steady-state spectroscopic results.

**Table 2 Time constants determined from TA measurements for *Anti*-DPyB and *Syn*-DPyB in *n*-hexane and acetonitrile[a].**

| | | $\tau_1$ (ps) | $\tau_2$ (ps) | $\tau_3$ (ps) | $\tau_4$[a] (ns) | $\tau_5$[a] (ns) |
|---|---|---|---|---|---|---|
| *Anti*-DPyB | In *n*-hexane | 3.6 ± 0.3 | 231 ± 19 | 1750 ± 116 | >10 | >10 |
| | In acetonitrile | 2.8 ± 0.1 | 24.3 ± 0.5 | 495.7 ± 6.5 | >10 | >10 |
| *Syn*-DPyB | In *n*-hexane | 2.3 ± 0.8 | – | 9.7 ± 0.5 | 6.4 ± 0.2 | – |
| | In acetonitrile | 2.8 | – | 8.0 ± 0.6 | 4.8 ± 0.2 | – |

[a]The $(T_1T_1)$ has two fates: the decay to the ground state ($\tau_4$) and the dissociation to free triplets ($\tau_5$). We could not distinguish two process because the lifetime of $(T_1T_1)$ is longer than the investigated delay times. Thus we denote two time constants for the decay to the ground state and the dissociation to free triplets as >10 ns.

**Table 3 Electrochemical parameters of Py, *Anti*-DPyB and *Syn*-DPyB evaluated by cyclic voltammograms ($E_{ox1}$ and $E_{ox2}$: first and second oxidation potentials, $E_{red1}$ and $E_{red2}$: first and second reduction potentials).**

| | $E_{ox1}$[a,b] (V) | $E_{ox2}$[a,b] (V) | $E_{red1}$[a,c] (V) | $E_{red2}$[a,c] (V) | $E_{ox2}-E_{ox1}$ (V) | $E_{red2}-E_{red2}$ (V) |
|---|---|---|---|---|---|---|
| Py | 0.85 | | −2.22 | | | |
| *Anti*-DPyB | 0.83 | 0.98 | −2.07 | −2.16 | 0.15 | 0.09 |
| *Syn*-DPyB | 0.85 | 1.12 | −2.06 | −2.18 | 0.27 | 0.12 |

[a]Determined by cyclic voltammetry (vs. SCE).
[b]Measured in $CH_2Cl_2$.
[c]Measured in THF.

We note that the excimer formation in *Anti*-DPyB and *Syn*-DPyB is intramolecular. The emission spectra of *Anti*-DPyB and *Syn*-DPyB are not influenced by the solute concentration (Supplementary Fig. S16). This result confirms that the excimers in *Anti*-DPyB and *Syn*-DPyB form due to intramolecular rather than intermolecular interaction. Because, in *Anti*-DPyB, the interaction between the two Py moieties in the ground state is weak due to the long distance between the two chromophores with twisted alignment, the intramolecular excimer formation requires the rearrangement of two distant Py moieties. This scenario is consistent with the flexible structure of the excimer of *Anti*-DPyB, reflected in its broad structureless excimer emission band (~480 nm). In contrast, *Syn*-DPyB is expected to rapidly form an excimer with no or less structural rearrangement because of the partial overlap of the two Py moieties. Based on the structured excimer emission band (~480 nm) of *Syn*-DPyB, we suggest that the excimer structure is as rigid as the structure in the ground state.

The solvent dependence on the emission spectra of *Anti*-DPyB can be rationalized by considering the following scenario: As Py is a hydrophobic molecule, two Py moieties in a nonpolar solvent show monomeric behavior, whereas a high polarity solvent facilitates the hydrophobic interaction of two Py moieties, resulting in more efficient excimer formation. The absence of solvent dependence on the emission of *Syn*-DPyB is probably due to its rigid structure owing to the strong π–π interaction between two Py moieties. The solvent dependency on the emission of *Anti*-DPyB and *Syn*-DPyB can be also interpreted in terms of the change in the dipole moment ($\mu$) induced by the structural change in the excited state. The solvent dependency on the emission of *Anti*-DPyB suggests that the $\Delta\mu$ ($=\mu_e-\mu_g$) value for *Anti*-DPyB is likely larger than that for *Syn*-DPyB. The DFT calculation, which was performed using CAM-B3LYP-D3/6-31G**, shows that the dipole moments ($\mu_g$) of *Anti*-DPyB and *Syn*-DPyB in the ground state are 0.021 and 0.1394 D, respectively (see Supplementary Data 2). According to the TDDFT calculation, the dipole moments ($\mu_e$) of the optimized *Anti*-DPyB and *Syn*-DPyB in the excited state are 0.3052 and 0.2729 D, respectively. These TDDFT/DFT calculations demonstrate that the $\Delta\mu$ (0.284 D) of *Anti*-DPyB is larger than that of *Syn*-DPyB (0.134 D), suggesting that in the excited state, the structural change in *Anti*-DPyB to form the excimer may induce a relatively large $\Delta\mu$ compared to that of *Syn*-DPyB. Unlike *Anti*-DPyB and

*Syn*-DPyB that form the excimer in the excited state, 1,4-di(1-pyrenyl)benzene (Py-Benz-Py), which is a covalently *para*-linked pyrene (Py) dimer, shows significant different excited-state relaxation dynamics[9]. Indeed, it was reported that Py-Benz-Py exhibits solvent-dependent ICT dynamics, followed by the twisting motion between Py and phenyl moieties, without intramolecular excimer formation[9]. The difference in the excited-state relaxation dynamics of *Anti*-DPyB, *Syn*-DPyB, and Py-Benz-Py indicates that the molecular structure and configuration play a vital role in their excited-state relaxation dynamics.

**Excited-state dynamics dependent on molecular configuration.** As shown in Table 2, the TA data for both *Anti*-DPyB and *Syn*-DPyB show similar $\tau_1$ time constants (2.3–3.7 ps) regardless of solvent polarity. This time scale falls into the well-known time scale for vibrational relaxation. Thus, the earliest kinetic component ($\tau_1$) of ~3 ps can be interpreted as the intramolecular vibrational relaxation (IVR) from the initially populated local excited state (Franck–Condon state).

After IVR ($\tau_1$), the excited molecules in the S1 state have various potential fates, including relaxation to other excited states, such as excimer or triplet excited states, and returning to the ground state via fluorescence. The observation of excimer fluorescence for *Anti*-DPyB and *Syn*-DPyB leads to the interpretation that a part of the excited molecules in the S1 state relaxes to the excimer state. As discussed above, the excimer formation in *Anti*-DPyB should require the rearrangement of the two distant Py moieties to induce the interaction between them. In this regard, the $\tau_2$ time constants observed from *Anti*-DPyB can be interpreted as conformational change. The $\tau_2$ time constant (231 ps) of *Anti*-DPyB in *n*-hexane is similar to the 323 ps assigned to the twisting motion between the Py and phenyl moiety of Py-Benz-Py[9] in a nonpolar solvent. Thus, we attribute the $\tau_2$ of *Anti*-DPyB to excimer formation via S1 → excimer transition accompanying the twisting motion between the Py and phenyl moieties. The faster excimer formation in acetonitrile probably occurs due to the strong hydrophobic interaction between the two Py moieties in a high-polarity solvent. The excimer of *Anti*-DPyB decays to a long-lived species with $\tau_3$ time constants. The TA spectra of the long-lived species are highly similar to the triplet–triplet absorption spectra of free Py derivatives corresponding to the $T_1 \rightarrow T_n$ transition[48,49].

Specifically, the long-lived species of *Anti*-DPyB show structured TA spectra, similar to the $T_1$-to-$T_n$ absorption spectra for carbonylpyrenes reported by Rajagopal et al.[49]. Furthermore, the TA signals of *Anti*-DPyB observed at >5 ns resemble the $T_1$-to-$T_n$ absorption spectrum of 1-(2-bromophenyl)pyrene measured in dichloromethane (DCM) (Supplementary Fig. S17), although the peak positions are slightly different from each other. Therefore, the long-lived species observed in *Anti*-DPyB are $T_1$ (free triplet state) or a similar state that produces an absorption spectrum similar to that of $T_1$-to-$T_n$.

The second fastest time constant (9.7 and 8.0 ps in *n*-hexane and acetonitrile, respectively) for *Syn*-DPyB is faster than that for *Anti*-DPyB by two orders of magnitude. Nevertheless, these time scales are much longer than the typical subpicosecond time scale reported for the excimer formation of prestacked dimeric systems[24,30]. For example, Hong et al. showed that the excimer state of cofacial stacked perylene bisimide dimer is formed within 200 fs[24]. Unlike *Anti*-DPyB, the excimer in *Syn*-DPyB with a prestacked structure should be rapidly formed with no or less structural rearrangement. For this reason, we ruled out the possibility that the second fastest time constants in *Syn*-DPyB can be attributed to excimer formation dynamics. Instead, we consider two possibilities: the excited molecules in $S_1$ formed by IVR (~3 ps) relax to the excimer state (i) within a subpicosecond (≤200 fs) and (ii) with a time constant comparable to IVR (~3 ps). Notably, the TA spectra of a long-lived species formed with the second fastest time constants in *Syn*-DPyB are highly similar to the triplet–triplet absorption spectra of free Py derivatives corresponding to the $T_1 \rightarrow T_n$ transition, like those of a long-lived species formed with $\tau_3$ time constants in *Anti*-DPyB. For this reason, we denote the time constants of 9.7 and 8.0 ps for *Syn*-DPyB as $\tau_3$, not $\tau_2$. In other words, the long-lived species in *Anti*-DPyB and *Syn*-DPyB are formed with $\tau_3$ time constants and they are attributed to $T_1$ or a similar state.

The free triplet states of a molecule can be generated through an ISC process or the dissociation of $(T_1T_1)$. It has been accepted that the ISC process in organic molecules with a small spin–orbit coupling occurs with a timescale of 10 ns to 1 ms and the lifetime of the triplet state is longer than the timescale in the order of 1 μs. In contrast to ISC, $(T_1T_1)$ rapidly forms within the range of 10 fs to 1 ns and has a significantly shorter lifetime than that of the free triplet formed via the ISC process, although the TA spectrum for the $(T_1T_1)$ is similar to that of the free triplet formed through ISC. A previous study of Py in micelles reported an ISC time constant of 1.7 μs[50], which is much longer than the $\tau_3$ time constants of 1.75 ns and 495.7 ps for the formation of the long-lived species of *Anti*-DPyB in *n*-hexane and acetonitrile, respectively. Meanwhile, the long-lived species for *Syn*-DPyB in *n*-hexane and acetonitrile are formed with $\tau_3$ time constants of 9.7 and 8.0 ps, respectively, and then relax to other states with $\tau_4$ time constants of 6.4 and 4.8 ns, respectively. These lifetimes of the long-lived species of *Anti*-DPyB and *Syn*-DPyB are significantly shorter than the triplet lifetimes of Py (9.4–11 ms)[51]. Thus, we attribute the long-lived species observed from *Anti*-DPyB and *Syn*-DPyB to the $(T_1T_1)$ formed through the first step of SF. Similar examples were reported for the SF dynamics of bis(triisopropylsilylethynyl)-tetracene (Tips-tetracene) in solution, which occur via an excimer with a $(T_1T_1)$ character[34], and four 3,6-bis(thiophene-2-yl)diketopyrrolopyrrole derivatives with different substituents in film, which involve an intermediate excimer-like state[52]. However, it has been generally accepted that excimer formation inhibits an SF process[30,38]. Contrary to this generally accepted view, our results demonstrate that the excimers in both *Anti*-DPyB and *Syn*-DPyB are rapidly converted to $(T_1T_1)$. $2E_{T1}$ values of *Anti*-DPyB (4.08 eV) and *Syn*-DPyB (3.74 eV) are higher than their $E_{S1}$ values (3.3 and 3.1 eV, respectively), suggesting that the SF processes in *Anti*-DPyB and *Syn*-DPyB are endothermic reactions.

**Singlet fission.** The dissociation dynamics of $(T_1T_1)$ in the SF process is key to determining the energy conversion efficiency in photoelectric or electrochemical devices, as the faster dissociation of $(T_1T_1)$ to free triplets is preferable in terms of energy conversion efficiency. The nanosecond TA experiments for *Anti*-DPyB and *Syn*-DPyB provide a clue for the free triplet that can be generated by the dissociation dynamics of $(T_1T_1)$. The nanosecond TA spectra for *Anti*-DpyB and *Syn*-DPyB suggest that the triplet species of *Anti*-DPyB exists at the μs–ms time scale, whereas *Syn*-DPyB does not exhibit any absorption band in both *n*-hexane and acetonitrile at this time scale. As shown in Supplementary Figs. S14 and S18, the absorption band around 445 nm of *Anti*-DPyB measured from the nanosecond TA experiment is similar to the $T_1$-to-$T_n$ absorption spectrum of 1-(2-bromophenyl)pyrene measured in DCM (Supplementary Fig. S17), suggesting that the chemical species of *Anti*-DPyB observed at the μs–ms time scale are attributed to $T_1$. To further confirm our interpretation, we additionally performed the nanosecond TA experiment for *Anti*-DPyB in iodomethane to maximize the heavy atom effect. As shown in Supplementary Fig. S18A, *Anti*-DPyB in iodomethane exhibits an intense absorption band in the range of 350–600 nm at a 1 μs time delay, which is almost identical to those of *Anti*-DPyB measured in *n*-hexane and acetonitrile using femtosecond TA spectroscopy. Thus, we suggest that the chemical species of *Anti*-DPyB observed at the μs–ms time scale are the free triplets generated by the dissociation dynamics of $(T_1T_1)$. Furthermore, we performed the nanosecond TA experiment for Py, Ph-Py, and *Syn*-DPyB in iodomethane. Their TA spectra show absorption bands in the range of 350–600 nm at a 1 μs time delay (Supplementary Fig. S18B). The absorption band of *Syn*-DPyB measured in *n*-hexane and acetonitrile using femtosecond TA spectroscopy is similar to the $T_1$-to-$T_n$ absorption band of *Syn*-PDyB measured in iodomethane. In this regard, the last relaxation times ($\tau_4$) observed from *Anti*-DPyB and *Syn*-DPyB by femtosecond TA experiments are attributed to the dissociation dynamics of $(T_1T_1)$.

On the other hand, the $(T_1T_1)$ has two fates: (i) dissociation to free triplets and (ii) decay to the ground state (Fig. 4). The femtosecond TA measurements showed that the time profile for transient absorption bands of *Anti*-DPyB around 440 nm, which well reflects the relaxation kinetics of $(T_1T_1)$, shows slow but distinct rising features (Supplementary Fig. S19A), whereas *Syn*-DPyB shows a relatively fast relaxtion dynamics of $(T_1T_1)$ to $2T_1$ and $S_0$ in parallel with a few nanosecond time constants (Supplementary

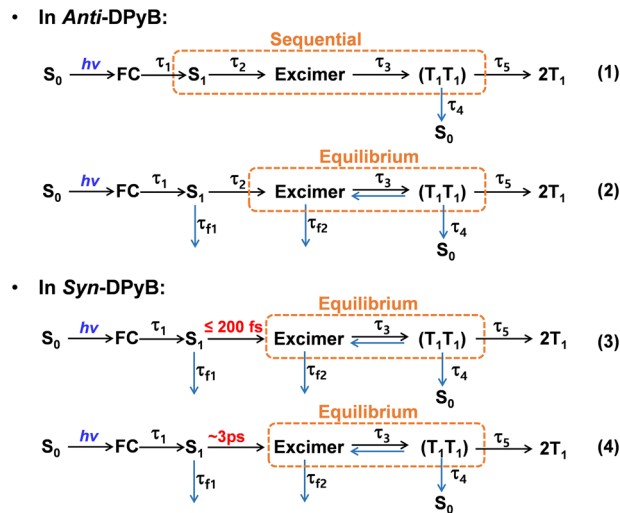

**Fig. 4 Kinetic schemes for photoinduced reactions of *Anti*-DPyB and *Syn*-DPyB.** $S_0$: ground state, FC: Franck–Condon state, $S_1$: singlet excited state, $(T_1T_1)$: correlated triplet pair, and $T_1$: free triplet state.

Fig. S19B and Table 2). This difference in the time profiles of *Anti*-DPyB and *Syn*-DPyB suggests that the $(T_1T_1)$ of *Anti*-DPyB dissociates dominantly into $2T_1$, whereas the $(T_1T_1)$ of *Syn*-DPyB decays mainly to $S_0$ state. Using the data from nanosecond TA spectroscopy, the triplet quantum yield ($\Phi_T$) values of *Anti*-DPyB in *n*-hexane and acetonitrile are determined to be 44.1% and 17.5%, respectively (Supplementary Methods). Meanwhile, we could not determine the $\Phi_T$ values of *Syn*-DPyB in *n*-hexane and acetonitrile, suggesting that the dissociation reaction of $(T_1T_1)$ in *Syn*-DPyB is significantly suppressed compared to *Anti*-DPyB and the $(T_1T_1)$ of *Syn*-DPyB decays mainly to $S_0$ state. The suppressed dissociation of $(T_1T_1)$ into free triplets in *Syn*-DPyB is probably due to triplet–triplet annihilation (TTA), which can be facilitated by the proximity of two Py moieties in the excimer state. Several studies of SF suggested that TTA is one of the decay processes of $(T_1T_1)$[17,53–55]. The TTA results in two fates: decay to the ground state and upconversion to a higher excited singlet state (excimer). The suppressed dissociation of $(T_1T_1)$ into free triplets in *Syn*-DPyB is probably due to TTA. Since the upconversion to the excimer state proceeds on a time scale in the tens of picoseconds, we attribute the $\tau_4$ of *Syn*-DPyB to the $(T_1T_1) \rightarrow S_0$ relaxation dynamics.

To confirm the SF dynamics, we performed time-resolved electron paramagnetic resonance (TR-EPR) measurements for *Anti*-DPyB and *Syn*-DPyB. The X-band (9.728 GHz) perpendicular mode TR-EPR spectra of *Anti*-DPyB and *Syn*-DPyB in toluene were measured at 80 K. Supplementary Fig. S20 shows EPR spectra of *Anti*-DPyB and *Syn*-DPyB at 128 and 200 ns after photoirradiation. The EPR signals for *Anti*-DPyB and *Syn*-DPyB show the narrow peak splitting of 34 and 19 mT around 340 mT ($g = 2.002$), respectively. In addition to the narrow peak splitting, *Anti*-DPyB and *Syn*-DPyB exhibit a large peak splitting of 150 and 115 mT, respectively. The EPR signals for *Anti*-DPyB and *Syn*-DPyB are well reproduced by the simulated curve for its triplet (see Supplementary Information). We further confirmed the origins of TR-EPR signals via the nutation experiment for the Q-band (34 GHz) TR-EPR signal of *Anti*-DPyB (see Supplementary Fig. S21). Although the nutation measurement on the EPR signal of *Syn*-DPyB was not performed, we speculate that the X-band EPR signal measured from *Syn*-DPyB arises from triplet species as well. The EPR signals at a few hundred nanoseconds do not show evidence for $(T_1T_1)$. The absence of EPR signals of $(T_1T_1)$ for *Anti*-DPyB and *Syn*-DPyB at a few hundred nanoseconds is probably due to the shorter lifetimes of $(T_1T_1)$s than the temporal resolution (~120 ns) of our TR-EPR system. *Syn*-DPyB shows a relatively fast decay feature in the time profile for TA bands of 450 nm (Supplementary Fig. S19B). As shown in Table 2, the $(T_1T_1)$s of *Syn*-DPyB in *n*-hexane and acetonitrile relax to $2T_1$ and $S_0$ in parallel with time constants of 6.4 and 4.8 ns, respectively, indicating that the lifetime of $(T_1T_1)$ for *Syn*-DPyB should be significantly shorter than the temporal resolution (~120 ns) of our TR-EPR system. Meanwhile, we could not precisely determine the lifetime of $(T_1T_1)$ for *Anti*-DPyB because of the limited range of investigated delay times in the femtosecond TA measurement. Overall, the EPR data lead us to conclude that the lifetime of $(T_1T_1)$ for *Anti*-DPyB should be shorter than the temporal resolution (~120 ns) of our TR-EPR system.

The difference in the SF dynamics of *Anti*-DPyB and *Syn*-DPyB may be interpreted in terms of the $E_b$ of $(T_1T_1)$. According to the kinetic model for SF proposed by Kolomeisly et al., the rates of the formation and dissociation of $(T_1T_1)$ depend on $E_b$[56]. They defined that if $E_b > 0$, then the $(T_1T_1)$ state is bound, and if $E_b < 0$, then it is unbound. The $(T_1T_1)$ destabilization ($E_b < 0$) simultaneously results in the fast dissociation of $(T_1T_1)$ and the slow formation of $(T_1T_1)$. Based on a spin-lattice model, Abraham and Mayhall predicted that in various covalently linked tetracene or pentacene dimers[42], the $(T_1T_1)$ state of the *meta*-

linked dimer would be unbound with respect to the separated triplets due to the smaller or negative $E_b$ compared to those of the *ortho*- and *para*-linked dimers, whereas the $(T_1T_1)$ states of the *ortho*- and *para*-linked dimers would be bound. Similarly, the analysis of wave functions by Chesler et al. showed that the slow formation of $(T_1T_1)$ in *meta*-bianthracene may be due to the small or negative $E_b$, whereas the fast formation of $(T_1T_1)$ of *para*-bianthracene results from the larger or positive $E_b$[41]. In other words, these theoretical studies predicted that compared to *ortho*- and *para*-linked dimers, *meta*-linked dimers with a smaller $E_b$ would exhibit a relatively slower formation of $(T_1T_1)$ state and a relatively fast dissociation of $(T_1T_1)$ into free triplets with a large yield. Indeed, the experimental results for several CLDs agree with the theoretical predictions[29,41–43]. The theoretical calculation results for many *ortho*-linked CLDs predict that *Anti*-DPyB and *Syn*-DPyB, which are also *ortho*-linked CLDs, form the bound $(T_1T_1)$ state due to the large or positive $E_b$. The TA data demonstrate that *Syn*-DPyB shows a low dissociation reaction of the $(T_1T_1)$ into free triplets, suggesting that the $(T_1T_1)$ state of *Syn*-DPyB is relatively bound compared to that of *Anti*-DPyB. In contrast to *Syn*-DPyB, *Anti*-DPyB shows a significantly slower formation of $(T_1T_1)$ followed by the dissociation into free triplets, suggesting that the $(T_1T_1)$ state of *Anti*-DPyB is unbound. This result indicates that as with *meta*-linked dimers that show favorable SF dynamics, *Anti*-DPyB, even if it is an *ortho*-linked dimer, forms the unbound $(T_1T_1)$ state due to a small or negative $E_b$, leading to the efficient dissociation of $(T_1T_1)$ into free triplets. The theoretical calculation results by Nakano and coworkers demonstrated that compared to *ortho*- and *para*-linked pentacene dimers, the electronic coupling between chromophores for the *meta*-linked pentacene dimer is very low, resulting in the relatively slow formation of $(T_1T_1)$ and an efficient SF[57,58]. In this regard, the efficient SF in *Anti*-DPyB is due to the low electronic coupling owing to the twisted alignment of the two chromophores. This result indicates that the SF dynamics in *ortho*-linked dimers, which show a significant π-orbital overlap between two chromophores, can be modulated by controlling the molecular configuration. Consequently, our results for *Anti*-DPyB and *Syn*-DPyB suggest that the molecular geometry of a CLD plays a critical role in their SF dynamics as well as excimer formation and ICT.

**TA spectra analysis with kinetic models**. We performed the kinetic analysis of the TA spectra of *Anti*-DPyB and *Syn*-DPyB considering various plausible kinetic models. For *Anti*-DPyB, considering five principal components from SVD analysis (Supplementary Fig. S11) and four-time constants obtained from the fitting of rSVs (Supplementary Fig. S13), we set up the simplest kinetic model with five intermediates assigned to FC, $S_1$, excimer, $(T_1T_1)$, and $2T_1$, and the four-time constants connecting them. Considering the relaxation process of the $(T_1T_1)$ state into $2T_1$ and $S_0$, the decay process from $(T_1T_1)$ to $S_0$ was added to the kinetic model. The resulting kinetic model is Kinetic Model (1) in Fig. 4. Details regarding Kinetic Model (1) are provided in Supplementary Information.

Whereas Kinetic Model (1) could explain the TA data well, it could not explain the emission behavior. The fluorescence decay profiles showed two-time constants assigned to the fluorescence lifetimes of the Py monomeric unit and excimer. By adding these two fluorescence decay times to Kinetic Model (1), we set up a different kinetic model (Kinetic Model (2) in Fig. 4). In this kinetic model, we also included the backreaction from $(T_1T_1)$ to the excimer for the following reason: As shown in the inset of Fig. 2A, the rising time of 1.24 ns in the fluorescence decay profile for *Anti*-DPyB in *n*-hexane is approximately five times larger than the time

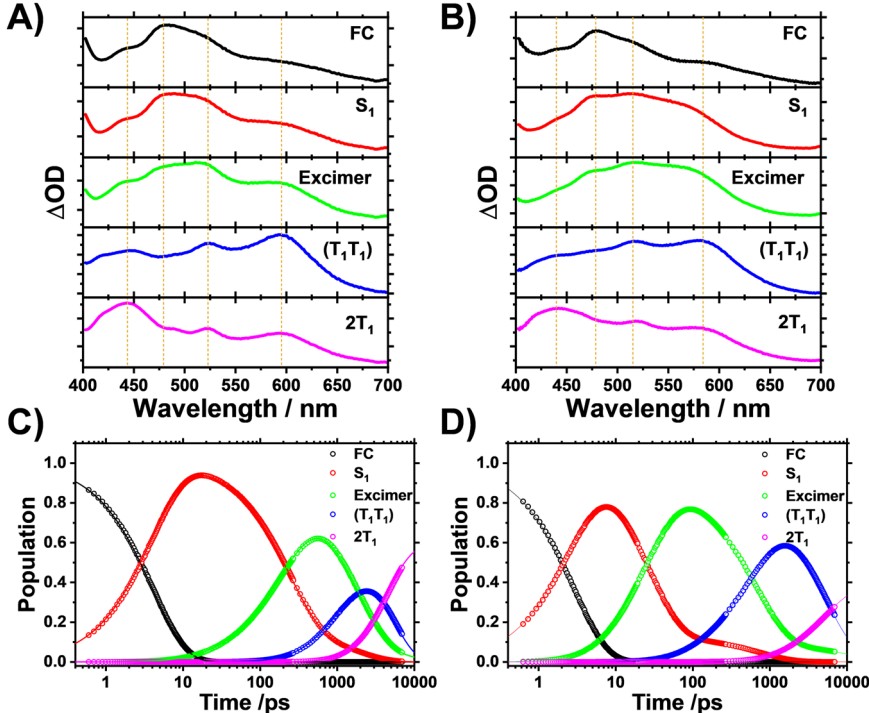

**Fig. 5 Species-associated difference spectra and population changes of intermediates for *Anti*-DPyB obtained from the kinetic analysis for Kinetic Model (2) in *n*-hexane and acetonitrile. A**, **B** Species-associated difference spectra in **A** *n*-hexane and **B** acetonitrile. **C**, **D** Population changes of intermediates in **C** *n*-hexane and **D** acetonitrile. The solid lines are the concentrations obtained from the kinetics analysis. The open circles represent the measure time delays.

constant (0.23 ns) corresponding to the $S_1 \rightarrow$ excimer transition determined from femtosecond TA experiments. This difference indicates that the observed excimer fluorescence is not prompt emission but delayed emission, suggesting equilibrium between the excimer state and the $(T_1T_1)$ state. Details regarding Kinetic Model (2) are provided in Supplementary Information. Figure 5 shows the five SADS curves and population changes for five intermediates (FC, $S_1$, excimer, $(T_1T_1)$, and $2T_1$) obtained from the kinetic analysis for Kinetic Model (2), and Supplementary Fig. S24 shows the experimental TA spectra, the best-fit spectra, and the residuals between them for *Anti*-DPyB in *n*-hexane and acetonitrile obtained from the kinetic analysis using Kinetic Model (2). The residuals between the experimental and the best-fit spectra are small, suggesting that the measured TA spectra for *Anti*-DPyB are well constructed as a linear combination of the five SADS curves according to the employed kinetic model. As the fit qualities of both Kinetic Models (1) and (2) are comparable, fit qualities alone could not be used to determine which kinetic model is more accurate (Supplementary Figs. S22–S24). As discussed above, however, Kinetic Model (2) is preferred because it is more consistent with the emission data.

For *Syn*-DPyB, considering the three exponential time constants obtained from the exponentials fitting of rSVs (Supplementary Fig. S13), the two-time constants and emission quantum yields from the emission experiments (Table 1), and the four principal components from SVD analysis results (Supplementary Fig. S12), we can preferentially set up a kinetic model with five-time constants and four intermediates. Simultaneously, the TR-EPR signal for *Syn*-DpyB indicates that the $(T_1T_1)$ of *Syn*-DPyB also dissociates to free triplets. Based on this result, the $(T_1T_1) \rightarrow 2T_1$ transition was included in the kinetic model. We also included the backreaction from $(T_1T_1)$ to the excimer as in *Anti*-DPyB. Consequently, as in *Anti*-DPyB, we used the kinetic model with five intermediates assigned to FC, $S_1$, excimer, $(T_1T_1)$, and $2T_1$

(see Fig. 4). In the case of *Syn*-DPyB, it is noteworthy that the $\tau_2$ time constant corresponding to the $S_1 \rightarrow$ excimer transition was not observed in *Syn*-DPyB (Table 2). The excimer of *Syn*-DPyB with a pre-stacked structure likely forms quickly within a subpicosecond ($\leq 200$ fs) or with a time constant comparable to the IVR (~3 ps). In this regard, we considered two kinetic models (Kinetic Models (3) and (4) in Fig. 4). In the former kinetic model, the $S_1 \rightarrow$ excimer transition occurs in a subpicosecond ($\leq 200$ fs), and in the other kinetic model, the $S_1 \rightarrow$ excimer transition occurs with a time constant comparable to the IVR (~3 ps). As shown in Supplementary Figs. S25 and S26, both Kinetic Models (3) and (4) show small residuals between the experimental and the best-fit spectra, suggesting that the measured TA spectra for *Syn*-DPyB are well constructed as linear combinations of the five SADS curves according to both kinetic models, and that fit qualities alone cannot be used to determine which kinetic model is better. Nevertheless, the SADS curves from the two kinetic models are different and provide clues regarding which kinetic model is more accurate. Whereas the SADS for the $S_1$ state from Kinetic Model (4) is positive (Fig. 6), that from Kinetic Model (3) is strongly negative (Supplementary Fig. S25), which is not possible for excited state absorption (ESA) from the $S_1$ state, so Kinetic Model (3) could be ruled out. In other words, the kinetic analysis suggests that the $S_1 \rightarrow$ excimer transition in *Syn*-DPyB occurs with a time constant comparable to the IVR (~3 ps). Figure 6 shows the five SADS curves and population changes for five intermediates (FC, S1, excimer, $(T_1T_1)$, and $2T_1$) for *Syn*-DPyB obtained from the kinetic analysis for Kinetic Model (4) in *n*-hexane and acetonitrile. The small differences among SADS curves of *Syn*-DPyB are probably due to the rigid structure of *Syn*-DPyB in the ground and excited states and the small energy differences between the states.

The kinetic analyses demonstrate that in *Anti*-DPyB, the $(T_1T_1)$ formed through the excimer slowly dissociates into free triplets. On the other hand, it was also proposed that the intermolecular and

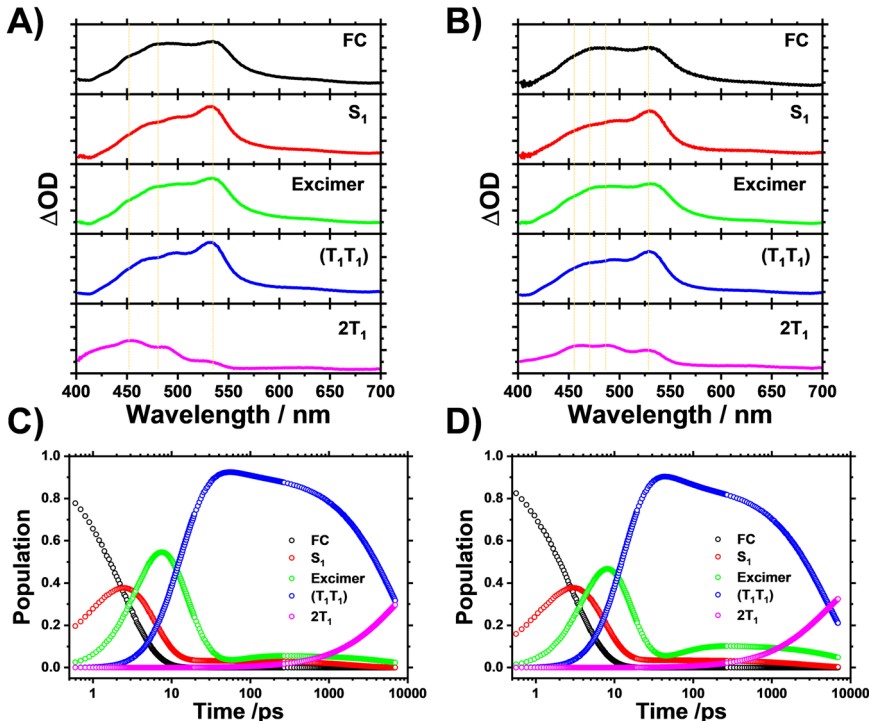

**Fig. 6 Species-associated difference spectra and population changes of intermediates for *Syn*-DPyB obtained from the kinetic analysis for Kinetic Model (4) in *n*-hexane and acetonitrile. A**, **B** Species-associated difference spectra in **A** *n*-hexane and **B** acetonitrile. **C**, **D** Population changes of intermediates in **C** *n*-hexane and **D** acetonitrile. The solid lines are the concentrations obtained from the kinetics analysis. The open circles represent the measure time delays.

intramolecular SF dynamics can rapidly occur with a direct process from the $S_1$ state to the free state due to strong coupling between the $S_1$ state and the free triplet state[59–61]. For example, Dover et al. suggested that the SF channel is dominated by a direct mechanism from the $S_1$ state and the formation of the excimer state inhibits efficient SF dynamics[61]. Thus, we also explored the possibility that our data from *Anti*-DPyB and *Syn*-DPyB can be explained using the same direct mechanism by applying the kinetic analysis with the reaction schemes compatible with the direct SF mechanism (see Supplementary Fig. S27). Those reaction schemes involving a direct SF process did not satisfactorily reproduce the measured TA spectra for *Anti*-DPyB in *n*-hexane and acetonitrile or yielded an unphysical SADS curve (Supplementary Information and Supplementary Fig. S28). This result indicates that the coupling between the $S_1$ state and the free triplet state in CLDs such as *Anti*-DPyB and *Syn*-DPyB is weaker than in the molecules that showed such direct SF processes, although further systematic studies are needed to confirm this hypothesis.

## Conclusions
To understand the ultrafast excited-state relaxation dynamics of intramolecular SF materials such as a CLD, we elucidated the ultrafast excited-state relaxation dynamics of covalently linked pyrene dimers *Anti*-DPyB and *Syn*-DPyB. In the excited state, *Anti*-DPyB, in which two Py moieties are oriented in a twisted configuration, forms the excimer through a conformational change with time constants of 231 and 24.3 ps in *n*-hexane and acetonitrile, respectively (Fig. 7). *Syn*-DPyB, with a pre-stacked configuration, rapidly forms the excimer without any conformational change with a time constant of ~3 ps. Our results also demonstrated that the excimer emissions observed from *Anti*-DPyB and *Syn*-DPyB are not prompt but delayed emissions. The time-resolved spectroscopic results showed that the resulting excimers rapidly relax to the $(T_1T_1)$ state, suggesting that the $(T_1T_1)$'s of *Anti*-DPyB and

*Syn*-DPyB are formed through the excimer state. The $(T_1T_1)$ of *Anti*-DPyB dissociates to free triplets as the end product, completing SF, whereas the dissociation reaction of $(T_1T_1)$ in *Syn*-DPyB is significantly suppressed compared to *Anti*-DPyB. This means that *Anti*-DPyB forms unbound $(T_1T_1)$, resulting in efficient SF dynamics, whereas *Syn*-DPyB forms bound $(T_1T_1)$. The suppressed dissociation of $(T_1T_1)$ into free triplets in *Syn*-DPyB is probably due to the triplet–triplet annihilation. This result differs from the prediction based on theoretical studies proposing that *meta*-linked dimers with a smaller $E_b$, compared with *ortho*- and *para*-linked dimers, exhibit efficient SF dynamics. This finding suggests that the efficiency of SF dynamics in CLDs cannot be predicted solely by the substitution position of the chromophore in a CLD. As *Anti*-DPyB and *Syn*-DPyB have relatively more distorted structures than previously studied CLDs (Fig. 1A), the orbital interaction likely has a much greater effect on their SF dynamics than the substitution position. Indeed, our data show that the relatively more efficient SF in *Anti*-DPyB compared to *Syn*-DPyB is caused by the relatively low electronic coupling between two chromophores owing to their twisted alignment. This result indicates that the SF dynamics in *ortho*-linked dimers, which generally show a significant π-orbital overlap between two chromophores, can be modulated by the control of the molecular configuration, consequently suggesting that the molecular geometry of a CLD plays a critical role in its SF dynamics, excimer formation, and ICT.

## Methods
**Synthetic procedures for *Anti*-DPyB and *Syn*-DPyB**. See Supplementary Methods and Supplementary Fig. S1 in the Supplementary Information.

**Characterization for *Anti*-DPyB and *Syn*-DPyB**. See Supplementary Figs. S2–S5 for [1]H- and [13]C-NMR spectra and Supplementary Fig. S6 for GC–MS data.

**Preparation of single crystal *Anti*-DPyB**. See Supplementary Methods.

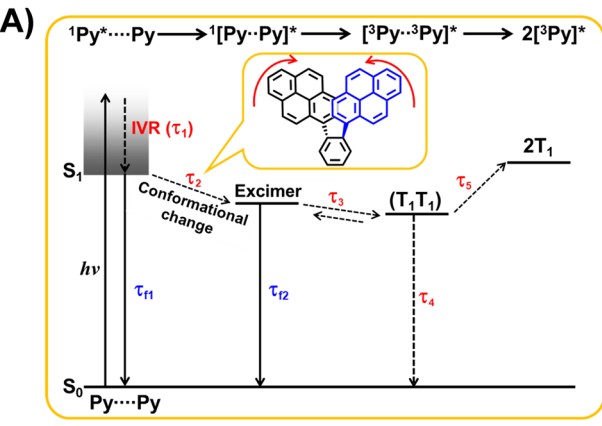

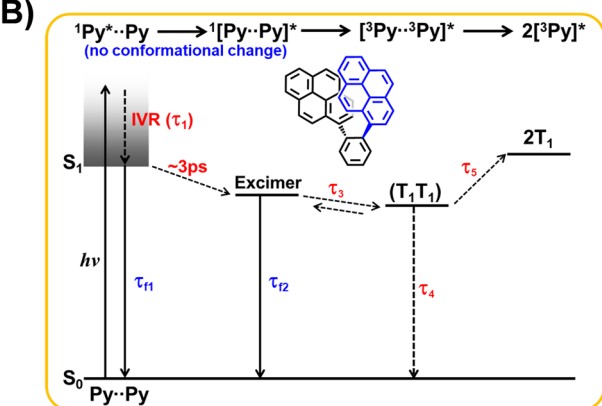

**Fig. 7 Schematics for proposed relaxation dynamics of *Anti*-DPyB and *Syn*-DPyB.** Excited-state relaxation dynamics of **A** *Anti*-DPyB and **B** *Syn*-DPyB. IVR stands for intramolecular vibrational relaxation.

**X-ray crystal structure analysis**. See Supplementary Methods, Supplementary Tables S1–S6, and Supplementary Fig. S7 for the thermal ellipsoid plot of *Anti*-DPyB.

**Steady-state and time-resolved spectroscopic measurements**. See Supplementary Methods, Supplementary Fig. S8 for fluorescence excitation spectra, Supplementary Fig. S9 for emission spectra of Py, Ph-Py, *Anti*-DPyB, and *Syn*-DPyB at 77 K, Supplementary Fig. S10 for femtosecond TA spectra of *Anti*-DPyB and *Syn*-DPyB, Supplementary Fig. S14 for nanosecond TA spectra of *Anti*-DPyB, Supplementary Fig. S16 for the concentration dependence of the emission spectra of *Anti*-DPyB and *Syn*-DPyB, Supplementary Fig. S17 for nanosecond TA spectra of 1-(2-bromophenyl)pyrene, Supplementary Fig. S18 for nanosecond TA spectra of Py, Ph-Py, *Anti*-DPyB, and *Syn*-DPyB, and Supplementary Fig. S19 for time profiles for transient absorption bands of *Anti*-DPyB and *Syn*-DPyB.

**Cyclic voltammograms of Py, *Anti*-DPyB, and *Syn*-DPyB**. See Supplementary Fig. S15.

**Triplet quantum yield (*Φ*$_T$) of *Anti*-DPyB**. See Supplementary Methods.

**Time-resolved EPR spectroscopy**. See Supplementary Methods and Supplementary Figs. S20 and S21.

**Singular value decomposition (SVD) analysis**. Details regarding SVD analysis are provided in the Supplementary Methods. The SVD analysis results (Supplementary Figs. S11 and S12) and the fits of rSVs (Supplementary Fig. S13) are provided in the Supplementary Methods.

**Kinetic analysis**. See Supplementary Methods. The results for the kinetic analysis of the TA spectra of *Anti*-DPyB and *Syn*-DPyB are given in Supplementary Figs. S22–S26.

**Direct SF mechanisms from the S$_1$ state**. See Supplementary Methods and Supplementary Figs. S27 and S28.

**Cartesian coordinates from computational studies**. See Supplementary Data 2.

## Data availability

All data generated during this study are all provided in the Article and its Supplementary Information, but are available from the authors upon reasonable request. Cartesian coordinates from computational studies can be found in Supplementary Data 2. The X-ray crystallographic coordinate for *Anti*-DPyB reported in this Article has been deposited at the Cambridge Crystallographic Data Centre (CCDC), under deposition number CCDC-2089494. This data can be obtained free of charge from The Cambridge Crystallographic Data Centre via www.ccdc.cam.ac.uk/data_request/cif. The cif file for *Anti*-DPyB can be found in Supplementary Data 1.

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

## Acknowledgements

This work was supported by the Institute for Basic Science (IBS-R033). This work was supported by the National Research Foundation of Korea (NRF), funded by the Ministry of Education (NRF-2020R1C1C1009007 and NRF-2017M3D1A1039380).

## Author contributions

J.C., K.-R.W., and H.I. designed research; J.C., S.K., M.A., J.K., D.W.C., D.K., S.E., D.I., Y.K., S.H.K., K.-R.W., and H.I. performed research; J.C., S.K., M.A., J.K., D.W.C., D.K., S.E., D.I., Y.K., S.H.K., K.-R.W., and H.I. contributed interpretation of results and J.C., K.-R.W., and H.I. wrote the paper.

## Competing interests

The authors declare no competing interests.
