## [Peer Review File · Communications Chemistry]

Singlet Fission Dynamics Modulated by Molecular Configuration in Covalently Linked Pyrene Dimers, Anti- and Syn-1,2-di(pyrenyl)benzeneReviewers' comments:

Reviewer #1 (Remarks to the Author):

The authors have several claims:

1. They claim that the conformational effect on the efficiency of iSF has not been addressed (Abstract: However, the effect of the molecular configuration on improving the efficiency through the spatial arrangement has not been explored yet.), which is overstated – here are a few examples randomly chosen from a substantial list of studies dedicated to iSF in CLDs where the conformation inevitably comes into the picture and is discussed either in terms of mechanism or dynamics or both: doi.org/10.1021/jacs.8b04830, doi.org/10.1039/C7CP03774K, doi.org/10.1002/anie.201802185, doi.org/10.1021/acs.jpcc.1c04734, doi.org/10.1021/acs.jpcb.0c08590.

2. They assert that ortho-linked chromophores with a conjugated spacer can exhibit iSF in a specific conformation, although in the literature meta-linked dimers are deemed the properly linked iSF CLDs. This conjecture is adequately demonstrated with abundant results from a number of experimental techniques.

3. They claim to have demonstrated the role of conformation on the SF propensity and to have deciphered the relaxation mechanism and revealed the reason of dissimilar behavior of the conformers. For the purpose, they have made detailed modelling, measurements and analysis of the excited state relaxation dynamics with a special focus on the role of the excimer formation in the iSF mechanism.

To prove their point, the authors have synthesized two conformers of o-dipyrenylbenzene – syn and anti. The synthesis is well documented and the obtained products are confirmed with elemental and GC-MS analyses, NMR spectra and X-ray diffraction compared to DFT calculations. The steady-state photophysical behavior of the compounds is rationalized based on absorption and emission spectra in solvents of different polarity. The excited-state relaxation dynamics is monitored by means of femtosecond transient absorption spectra subject to singular value decomposition analysis.

Nanosecond transient absorption spectra were taken to observe the long-lived triplets.

The assortment of experimental methods utilized gives certainty to the reader that the results are trustworthy. The analysis of the results and their interpretation is systematic and convincing, complemented by the well-organized tables and very good figures in the main text as well as the voluminous and informative Supporting materials. There is no need of additional experimental work. The conclusions are adequately substantiated.

A few minor comments, which may strengthen the authors' point could be:

- 1) A somewhat more exhaustive survey of the literature on the topic;
- 2) The suggestion for the different solvent effect on the emission intensities of the two conformers (p.6: In terms of solvent dependence, in Anti-DPyB, the relative intensities of the emissions from the monomeric Py moiety and the excimer show significant dependence on solvent polarity, whereas Syn-DPyB does not show definite solvent dependency.) could be supported by numerical values of the dipole moment from DFT calculations;
- 3) The use of τ_4 for two different channels of relaxation (Table 2, Scheme 1, Figure 6) is confusing; probably τ_5 could be introduced.

The topic of the paper and the obtained results are of interest to both the scientists working in the subfield and, more generally, to every researcher involved in the quest of materials for highly efficient organic photovoltaics.

The language is good, the size is appropriate, the readability is excellent.

This reviewer believes that the paper will attract the attention of the readership and recommends the manuscript for publishing without further reviewing.

Reviewer #2 (Remarks to the Author):

The manuscript entitled "Singlet Fission Dynamics Modulated by Molecular Configuration in Covalently Linked Pyrene Dimers, Anti- and Syn-1,2-di(pyrenyl)benzene" reports singlet fission dynamics in newly synthesized syn- and anti- pyrene dimers based on transient absorption measurements. We concern the core idea of this work as well as their experimental analysis. We politely ask the authors to consider our criticisms described below when they prepare the revised manuscript for another journal.

1) The following statement in the introduction section, "As can be seen in these examples, the results from numerous experimental and theoretical approaches for CLDs show diverse and inconsistent SF dynamics.", seems deviated from the current consensus on singlet fission dynamics and mechanisms which has been accomplished by numerous reports. Rather than unifying the mechanism of SF by a single model, its mechanism depends strongly on intermolecular interactions between SF active chromophores, which can be the direct coupling, charge-transfer mediation, etc. The authors statement may misguide future readers and seems to be exaggerated.

2) They proposed that the syn- and anti-pyrene dimers pose a valuable insights on SF dynamics. Though, as SF dynamics are very sensitive to intermolecular interactions, their structure are not ideal for the SF study in terms of elucidating SF mechanisms. Multiple components observed in the excited state may originate from structural heterogeneity of the molecules. Subtle differences in rotational angles and relative distances between two chromophores should result in a completely different excited-state dynamics. We recommend to check the cited references for the better understanding (Nat. Commun. 2016, 7, 13622 and J. Am. Chem. Soc. 2019, 141, 17558).

3) As pointed out from the second comment, the complicated excited-state dynamics in the proposed dimer should not be originated from their electronic states. Also, it seems not convincing to assign excited-state species regarding very similar TA spectra (FC vs Excimer). Authors should resolve this issue for the revised manuscript.

Reviewer #3 (Remarks to the Author):

The authors report the singlet fission dynamics of two different covalently-linked pyrene dimers. The number of SF systems using pyrene derivatives is extremely limited and the research concept may be relatively high. However, the experiment data already described in the text and supporting information are not sufficient to conclude the occurrence of singlet fission and its dynamics, and a significant amount of additional experiments are needed. In addition, a few important findings have been reported on the control of steric configuration for high-yield intramolecular SF and generation of free triplets using covalent-linked dimers, but they have not been fully explained in the introduction section. Authors should reconsider the research content following the reviewer's comments.

1. The degree of electronic coupling between two pyrene chromophores (Anti-DPyB vs. Syn-DPyB) should be estimated by absorption spectra and/or electrochemical method. If impossible, DFT calculations can provide some quantitative information.
2. Authors should measure phosphorescence spectra of these dimers or a reference monomer to experimentally demonstrate the triplet energies of pyrene derivatives in this study. Do these molecules satisfy the energy level matching condition between singlet and triplet excited states??
3. Although the authors propose a kinetic model as shown in Figure 6, the shape of the spectrum in each state is very similar, and the reliability of the relation to the proposed kinetic model is not sufficient.
4. Associated with above questions, authors should measure time-resolved EPR and directly assign the TT and T+T states, respectively.
5. Authors should carefully confirm the T-T absorption spectra of Anti-DPyB and Syn-DPyB by energy transfer from energy donor dye to these Pyrene dimers.
6. Associated with the above content, the molar absorption coefficients should be estimated. Then, the triplet yield of individual triplet excited states should be calculated.

7. Page 4 in the text:

Besides, the study on the effect of a molecular configuration, which can affect its excited-state relaxation dynamics, may provide a clue for the optimized spatial arrangements for the high energy conversion efficiency of a real device, but such studies are rare.

The reviewer does NOT think so. Authors should carefully check the recent progress regarding the molecular configuration of covalently-linked dimers.

8. Regarding the relationship between excimer state and singlet fission, authors should cite the following paper and discuss it.

Dover, C. B.; Gallaher, J. K.; Frazer, L.; Tapping, P. C.; Petty II, A. J.; Crossley, M. J.; Anthony, J. E.; Kee, T. W.; Schmidt, T. W., Endothermic singlet fission is hindered by excimer formation. *Nat. Chem.* 2018, 10, 305-310.

Responses to the comments from Reviewer #1

The authors have several claims:

1. They claim that the conformational effect on the efficiency of iSF has not been addressed (Abstract: However, the effect of the molecular configuration on improving the efficiency through the spatial arrangement has not been explored yet.), which is overstated – here are a few examples randomly chosen from a substantial list of studies dedicated to iSF in CLDs where the conformation inevitably comes into the picture and is discussed either in terms of mechanism or dynamics or both:

doi.org/10.1021/jacs.8b04830,

doi.org/10.1039/C7CP03774K,

doi.org/10.1002/anie.201802185,

doi.org/10.1021/acs.jpcc.1c04734,

[doi/10.1021/acs.jpcc.0c08590](https://doi.org/10.1021/acs.jpcc.0c08590).

→ We appreciate the valuable comments and suggestions on our study. We carefully read the papers recommended by the reviewer as well as related papers, which are now cited in the revised manuscript. Accordingly, we revised the abstract. The relevant part of the new abstract is as follows.

“Covalently linked dimers (CLDs) and their structural isomers have attracted much attention as potential materials for improving power conversion efficiencies through singlet fission (SF). Here, we designed and synthesized two covalently *ortho*-linked pyrene (Py) dimers, *anti*- and *syn*-1,2-di(pyrenyl)benzene (*Anti*-DPyB and *Syn*-DpyB, respectively), and investigated the effect of molecular configuration on SF dynamics using steady-state and time-resolved spectroscopies. [...]”

2. They assert that ortho-linked chromophores with a conjugated spacer can exhibit iSF in a specific conformation, although in the literature meta-linked dimers are deemed the properly linked iSF CLDs. This conjecture is adequately demonstrated with abundant results from a number of experimental techniques.

→ We appreciate the positive evaluation of our study.

3. They claim to have demonstrated the role of conformation on the SF propensity and to have deciphered the relaxation mechanism and revealed the reason of dissimilar behavior of the conformers. For the purpose, they have made detailed modelling, measurements and analysis of the excited state relaxation dynamics with a special focus on the role of the excimer formation in the iSF mechanism.

To prove their point, the authors have synthesized two conformers of o-dipyrenylbenzene – syn and anti. The synthesis is well documented and the obtained products are confirmed with elemental and GC-MS analyses, NMR spectra and X-ray diffraction compared to DFT calculations. The steady-state photophysical behavior of the compounds is rationalized based on absorption and emission spectra in solvents of different polarity. The excited-state relaxation dynamics is monitored by means of femtosecond transient absorption spectra subject to singular value decomposition analysis. Nanosecond transient absorption spectra were taken to observe the long-lived triplets.

The assortment of experimental methods utilized gives certainty to the reader that the results are trustworthy. The analysis of the results and their interpretation is systematic and convincing, complemented by the well-organized tables and very good figures in the main text as well as the voluminous and informative Supporting materials. There is no need of additional experimental work. The conclusions are adequately substantiated.

→ We appreciate the positive evaluation of our study. In the revised version, we tried to improve the manuscript by taking into account the comments of the reviewer as detailed in the following.

A few minor comments, which may strengthen the authors' point could be:

1) A somewhat more exhaustive survey of the literature on the topic;

→ We appreciate the reviewer's suggestion. Following that suggestion, we have thoroughly examined the literature on SF dynamics in CLDs and added the following additional references.

1. Chen M, Bae YJ, Mauck CM, Mandal A, Young RM, Wasielewski MR. Singlet Fission in Covalent Terrylenediimide Dimers: Probing the Nature of the Multiexciton State Using Femtosecond Mid-Infrared Spectroscopy. *J. Am. Chem. Soc.* **140**, 9184-9192 (2018). (Ref. #12 in the revised manuscript)
2. Dean JC, *et al.* Photophysical Characterization and Time-Resolved Spectroscopy of a Anthradithiophene Dimer: Exploring the Role of Conformation in Singlet Fission. *Phys. Chem. Chem. Phys.* **19**, 23162-23175 (2017). (Ref. #13 in the revised manuscript)
3. Yamakado T, Takahashi S, Watanabe K, Matsumoto Y, Osuka A, Saito S. Conformational Planarization versus Singlet Fission: Distinct Excited-State Dynamics of Cyclooctatetraene-Fused Acene Dimers. *Angew. Chem. Int. Ed.* **57**, 5438-5443 (2018). (Ref. #14 in the revised manuscript)
4. Nakamura S, *et al.* Synergetic Role of Conformational Flexibility and Electronic Coupling for Quantitative Intramolecular Singlet Fission. *J. Phys. Chem. C* **125**, 18287-18296 (2021). (Ref. #15 in the revised manuscript)
5. Paul S, Govind C, Karunakaran V. Planarity and Length of the Bridge Control Rate and Efficiency of Intramolecular Singlet Fission in Pentacene Dimers. *J. Phys. Chem. B* **125**, 231-239 (2021). (Ref. #16 in the revised manuscript)
6. Lukman S, *et al.* Tuning the Role of Charge-Transfer States in Intramolecular Singlet Exciton Fission through Side-Group Engineering. *Nat. Commun.* **7**, 13622 (2016). (Ref. #37 in the revised manuscript)
7. Alvertis AM, *et al.* Switching between Coherent and Incoherent Singlet Fission via Solvent-Induced Symmetry Breaking. *J. Am. Chem. Soc.* **141**, 17558-17570 (2019). (Ref. #26 in the revised manuscript)
8. Dover CB, *et al.* Endothermic Singlet Fission is Hindered by Excimer Formation. *Nat. Chem.* **10**, 305-310 (2018). (Ref. #61 in the revised manuscript)
9. Pun AB, *et al.* Ultra-Fast Intramolecular Singlet Fission to Persistent Multiexcitons by Molecular Design. *Nat. Chem.* **11**, 821-828 (2019). (Ref. #27 in the revised manuscript)
10. Fuemmeler EG, *et al.* A Direct Mechanism of Ultrafast Intramolecular Singlet Fission in Pentacene Dimers. *ACS Central Sci.* **2**, 316-324 (2016). (Ref. #59 in the revised manuscript)
11. Mauck CM, *et al.* Singlet Fission via an Excimer-Like Intermediate in 3,6-Bis(thiophen-2-yl)diketopyrrolopyrrole Derivatives. *J. Am. Chem. Soc.* **138**, 11749-11761 (2016). (Ref. #52 in the revised manuscript)
12. Fumanal M, Corminboeuf C. Direct, Mediated, and Delayed Intramolecular Singlet Fission Mechanism in Donor-Acceptor Copolymers. *J. Phys. Chem. Lett.* **11**, 9788-9794 (2020). (Ref. #60 in the revised manuscript)
13. Shizu K, Adachi C, Kaji H. Effect of Vibronic Coupling on Correlated Triplet Pair Formation in the Singlet Fission Process of Linked Tetracene Dimers. *J. Phys. Chem. A* **124**, 3641-3651 (2020) (Ref. #40 in the revised manuscript)

Some of the sentences with the references cited are as follows.

“Chromophore–chromophore interaction as well as the electronic-state coupling of a chromophore can modulate such electron carrier dynamics.⁷⁻¹⁶”

“In this regard, many multi-chromophore systems have been widely used for developing highly efficient photoelectric or electrochemical devices using chromophore–chromophore interaction.^{7, 12-21}”

“The results of extensive studies on SF dynamics and mechanisms showed that they cannot be easily explained by a single unifying mechanism of SF; instead, the intermolecular and intramolecular SF dynamics occur through various types of species such as charge transfer species, excimers, and higher excited vibrational and electronic states, depending on the interactions between chromophores.^{21, 24-26, 29, 31, 33-37}”

“Shizu et al. demonstrated that the efficient SF dynamics of *para*-bis(ethynyltetracenyl)benzene dimer is due to large vibronic coupling and the small energy difference between the singlet excited state and the ¹(T₁T₁) state.⁴⁰”

“Mauck et al. showed that the SF processes of four 3,6-bis(thiophene-2-yl)diketopyrrolopyrrole chromophores with different substituents in film involve an intermediate excimer-like state.⁵²”

“For example, Dover et al. suggested that the SF channel is dominated by a direct mechanism from the S₁ state and the formation of the excimer state inhibits efficient SF dynamics.⁶¹”

2) The suggestion for the different solvent effect on the emission intensities of the two conformers (p.6: In terms of solvent dependence, in Anti-DPyB, the relative intensities of the emissions from the monomeric Py moiety and the excimer show significant dependence on solvent polarity, whereas Syn-DPyB does not show definite solvent dependency.) could be supported by numerical values of the dipole moment from DFT calculations;

→ We appreciate the reviewer’s suggestion. Following the reviewer’s suggestion, we performed TDDFT/DFT calculations and added the following discussion to the revised text.

“The solvent dependence on the emission spectra of *Anti*-DPyB can be rationalized by considering the following scenario: As Py is a hydrophobic molecule, two Py moieties in a nonpolar solvent show monomeric behavior, whereas a high polarity solvent facilitates the interaction of two Py moieties, resulting in more efficient excimer formation. The absence of solvent dependence on the emission of *Syn*-DPyB is probably due to its rigid structure owing to the strong π – π interaction between two Py moieties. The solvent dependency on the emission of *Anti*-DPyB and *Syn*-DPyB can be interpreted in terms of the change in the dipole moment (μ) induced by the structural change in the excited state. The solvent dependency on the emission of *Anti*-DPyB suggests that the $\Delta\mu (= \mu_e - \mu_g)$ value for *Anti*-DPyB is likely larger than that for *Syn*-DPyB. To check this possibility, we calculated the μ values of *Anti*-DPyB and *Syn*-DPyB in the ground and excited states. The DFT calculation, which was performed using CAM-B3LYP-D3/6-31G**, shows that the dipole moments (μ_g) of *Anti*-DPyB and *Syn*-DPyB in the ground state are 0.021 and 0.1394 D, respectively. According to the TDDFT calculation, the dipole moments (μ_e) of the optimized *Anti*-DPyB and *Syn*-DPyB in the excited state are 0.3052 and 0.2729 D, respectively. These TDDFT/DFT calculations demonstrate that the $\Delta\mu$ (0.284 D) of *Anti*-DPyB is larger than that of *Syn*-DPyB (0.134 D), which are consistent with our hypothesis based on the spectroscopic results that *Anti*-DPyB forms the excimer through the rearrangement of the two distant Py moieties and *Syn*-DPyB rapidly forms the excimer with no or less structural rearrangement because of the partial overlap of the two Py moieties. In other words, in the excited state, the structural change in *Anti*-DPyB to form the excimer may induce a relatively large $\Delta\mu$ compared to that of *Syn*-DPyB.”

3) The use of τ_4 for two different channels of relaxation (Table 2, Scheme 1, Figure 6) is confusing; probably τ_5 could be introduced.

→ Considering the reviewer’s comment, we denote time constants corresponding to the $^1(T_1T_1) \rightarrow S_0$ relaxation and the dissociation of $^1(T_1T_1)$ as τ_4 and τ_5 , respectively. In this regard, we added the following new sentences in the revised manuscript.

“In this regard, the last relaxation time (>10 ns) observed from *Anti*-DPyB is attributed to the dissociation dynamics of $^1(T_1T_1)$, although we could not precisely determine this because of the limited range of investigated delay times. We denote the time constant corresponding to the dissociation dynamics of $^1(T_1T_1)$ as τ_5 .”

We also revised Table 2, Scheme 1, and Figure 6 as follows.

Table 2. Time constants determined from TA measurements for *Anti*-DPyB and *Syn*-DPyB in *n*-hexane and acetonitrile^a

		τ_1 / ps	τ_2 / ps	τ_3 / ps	τ_4^a / ns	τ_5^a / ns
Anti-DPyB	in n -hexane	3.6 ± 0.3	231 ± 19	1750 ± 116	> 10	> 10
	in acetonitrile	2.8 ± 0.1	24.3 ± 0.5	495.7 ± 6.5	> 10	> 10
Syn-DPyB	in n -hexane	2.3 ± 0.8	-	9.7 ± 0.5	6.4 ± 0.2	-
	in acetonitrile	2.8	-	8.0 ± 0.6	4.8 ± 0.2	-

^a The $^1(T_1T_1)$ has two fates: the decay to the ground state (τ_4) and the dissociation to free triplets (τ_5). We could not distinguish two process because the lifetime of $^1(T_1T_1)$ is longer than the investigated delay times. Thus we denote two time constants for the decay to the ground state and the dissociation to free triplets as > 10 ns.

In *Anti*-DPyB:

In *Syn*-DPyB:

Scheme 1. Kinetic schemes for photo-induced reactions of *Anti*-DPyB and *Syn*-DPyB. (S_0 : ground state, FC: Franck-Condon state, S_1 : singlet excited state, ${}^1(T_1T_1)$: correlated triplet pair, and T_1 : free triplet state).

Figure 6. Schematics for proposed relaxation dynamics of *Anti*-DPyB and *Syn*-DPyB. Excited-state relaxation dynamics of (A) *Anti*-DPyB and (B) *Syn*-DPyB. IVR stands for intramolecular vibrational relaxation.

The topic of the paper and the obtained results are of interest to both the scientists working in the subfield and, more generally, to every researcher involved in the quest of materials for highly efficient organic photovoltaics.

The language is good, the size is appropriate, the readability is excellent.

This reviewer believes that the paper will attract the attention of the readership and recommends the manuscript for publishing without further reviewing.

→ We appreciate the positive evaluation of our work.

Responses to the comments from Reviewer #2

The manuscript entitled “Singlet Fission Dynamics Modulated by Molecular Configuration in Covalently Linked Pyrene Dimers, Anti- and Syn-1,2-di(pyrenyl)benzene” reports singlet fission dynamics in newly synthesized syn- and anti- pyrene dimers based on transient absorption

measurements. We concern the core idea of this work as well as their experimental analysis. We politely ask the authors to consider our criticisms described below when they prepare the revised manuscript for another journal.

→ We appreciate the valuable comments and suggestions on our study. In the revised version, we tried to improve the manuscript by taking into account the comments of the reviewer as detailed in the following.

1) The following statement in the introduction section, “As can be seen in these examples, the results from numerous experimental and theoretical approaches for CLDs show diverse and inconsistent SF dynamics.”, seems deviated from the current consensus on singlet fission dynamics and mechanisms which has been accomplished by numerous reports. Rather than unifying the mechanism of SF by a single model, its mechanism depends strongly on intermolecular interactions between SF active chromophores, which can be the direct coupling, charge-transfer mediation, etc. The authors statement may misguide future readers and seems to be exaggerated.

→ We appreciate the valuable comments. As mentioned by the reviewer, the intermolecular and intramolecular SF dynamics occur through different types of species such as charge transfer species, excimers, and higher excited vibrational and electronic states, depending on interactions between chromophores. However, studies on SF dynamics of CLDs with similar molecular structures report at least seemingly contradictory results as mentioned in our previous manuscript. In this regard, in our original manuscript, we mentioned the following in the introduction section.

“As can be seen in these examples, the results from numerous experimental and theoretical approaches for CLDs show diverse and inconsistent SF dynamics.”

As the reviewer may consider that our explanation of the SF dynamics of CLDs appears a bit exaggerated, we revised the introduction as follows.

“The results of extensive studies on SF dynamics and mechanisms showed that they cannot be easily explained by a single unifying mechanism of SF; instead, the intermolecular and intramolecular SF dynamics occur through various types of species such as charge transfer species, excimers, and higher excited vibrational and electronic states, depending on the interactions between chromophores.^{21, 24, 25, 26, 29, 31, 33, 34, 35, 36, 37} For example, Zirzmeier et al. suggested that the SF process occurring in *ortho*-, *meta*-, and *para*-linked pentacene dimers proceeds through virtual CT states.²⁹ Margulies et al. reported that the covalently linked terylene-3,4:11,12-bis(dicarboximide) (TDI) dimer with a stacked structure forms an excimer within <200 fs, whereas the slip-stacked TDI dimer in a nonpolar solvent forms the correlated triplet pair (¹(T₁T₁)), which is an intermediate in the SF process.³⁰ They also showed that the slip-stacked TDI dimer forms a CT state in a few picoseconds in a polar solvent, suggesting that adjusting the CT state energy relative to exciton states via solute–solvent interaction can either promote or inhibit SF.³⁰ Ni et al. reported that, upon excitation at 250 nm, the cofacial perylene dimer undergoes SF from the upper excited vibrational and electronic states, whereas upon excitation at 450 nm, it fast forms an excimer, which relaxes to the ground state within nanoseconds.³⁸ Korovina et al. showed that the *ortho*- and *para*-bis(ethynyltetracenyl)benzene dimers with a relatively stronger through-bond coupling exhibit more efficient SF dynamics than the *meta*-bis(ethynyltetracenyl)-benzene dimer.³⁹ Especially, unlike the *ortho*-bis(ethynyltetracenyl)benzene dimer, which forms only ¹(T₁T₁), the *para*-bis(ethynyltetracenyl)benzene dimer shows complete SF dynamics to form free triplets, suggesting that the rotational flexibility between the acenes in the dimers plays an important role in SF dynamics.³⁹ Shizu et al. demonstrated that the efficient SF dynamics of *para*-bis(ethynyltetracenyl)benzene dimer is due to large vibronic coupling and the small energy difference between the singlet excited state and the ¹(T₁T₁) state.⁴⁰ In contrast, Nakamura et al. showed that in a series of *ortho*-, *meta*-, and *para*-

bis(triisopropylsilylethynyl)-tetracenyl)benzene dimers, the *meta*-linked tetracene dimer exhibits more efficient SF dynamics than *ortho*- and *para*-linked tetracene dimers.¹⁵ They suggested that the large conformational flexibility and weak electronic coupling play a critical role in their SF dynamics.¹⁵ In addition, the experimental and theoretical calculation results of various CLDs showed that compared to *ortho*- and *para*-linked dimers, *meta*-linked dimers exhibit more efficient SF dynamics due to the small binding energy (E_b) of $^1(T_1T_1)$ ($E_b = 2E|S_0T_1\rangle - E|^1T_1T_1\rangle$).^{29, 41, 42, 43}

Despite numerous experimental and theoretical approaches to determining the SF dynamics of CLDs, a full understanding of the parameters influencing the SF dynamics of CLDs is still lacking. Additionally, studies on the effects of a molecular configuration, which can affect its excited-state relaxation dynamics, may provide clues for the optimized spatial arrangements that ensure the high energy conversion efficiency of a real device. In this regard, the role of conformational flexibility in the SF dynamics of CLDs has been studied, but still needs further clarification. Accordingly, in-depth studies are needed to understand their excited-state relaxation dynamics, including SF.”

2) They proposed that the syn- and anti-pyrene dimers pose a valuable insights on SF dynamics. Though, as SF dynamics are very sensitive to intermolecular interactions, their structure are not ideal for the SF study in terms of elucidating SF mechanisms. Multiple components observed in the excited state may originate from structural heterogeneity of the molecules. Subtle differences in rotational angles and relative distances between two chromophores should result in a completely different excited-state dynamics. We recommend to check the cited references for the better understanding (Nat. Commun. 2016, 7, 13622 and J. Am. Chem. Soc. 2019, 141, 17558).

→ The reviewer speculated that multiple kinetic components for *Anti*-DPPyB and *Syn*-DPyB in the excited state are probably due to their structural heterogeneities. To check the existence of an intermolecular complex in the excited state, we evaluated the concentration effect on the emission spectra of *Anti*-DPyB and *Syn*-DPyB. As mentioned in our previous manuscript, the emission spectra of *Anti*-DPyB and *Syn*-DPyB are not influenced by the solute concentration. This result indicates that the intermolecular interaction in *Anti*-DPyB and *Syn*-DPyB is weak in our experimental condition and that the excimer formation in *Anti*-DPyB and *Syn*-DPyB is due to the intramolecular interaction. Furthermore, the structured emission spectrum of *Syn*-DPyB suggests that *Syn*-DPyB in the excited state has a rigid structure, suggesting there is no structural heterogeneity of *Syn*-DPyB in the excited state. Compared to *Syn*-DPyB, *Anti*-DPyB exhibited the broad structureless excimer emission band (~480 nm), indicating the flexible structure of the excimer of *Anti*-DPyB. However, we believe that the flexible structure of the excimer of *Anti*-DPyB does not induce the multiple kinetics in the excited state.

The excitation of 345 nm induces the transition from the ground state to a hot singlet excited state (Franck-Condon state). As explained in our previous manuscript, the steady-state spectroscopic results for *Anti*-DPyB and *Syn*-DPyB reveal formations of the singlet excited state (S_1) and excimer state. In addition, the time-resolved TA experiments for *Anti*-DPyB and *Syn*-DPyB suggest the existence of the long-lived species, the correlated triplet pair ($^1(T_1T_1)$). Thus, the excited molecules of FC state generated by photoexcitation rapidly relax to the singlet excited state (S_1). The kinetic analyses for the time-resolved TA spectra of *Anti*-DPyB and *Syn*-DPyB demonstrate that the excited-state relaxation dynamics of *Anti*-DPyB and *Syn*-DPyB are well described with a kinetic model (2) and (4), respectively.

On the other hand, SF dynamics mediated by a charge transfer species have been proposed in several CLDs. For example, two references recommended by the review also show that SF dynamics of 13,13'-bis(mesityl)-6,6'-dipentacenyl and 13,13'-bis((triisopropylsilyl)ethynyl)-6,6'-dipentacenyl proceed through a virtual charge transfer (CT) state and a real CT state, respectively. Furthermore, 1,4-

di(1-pyrenyl)benzene, which is a *para*-linked pyrene dimer, exhibits solvent-dependent ICT dynamics. Thus, we examined whether SF dynamics in *Anti*-DPyB and *Syn*-DPyB occur through CT state. As explained in our previous manuscript, *Anti*-DPyB and *Syn*-DPyB in *n*-hexane (non-polar solvent) and acetonitrile (high-polar solvent) show dual emission bands originating from Py monomer moiety and the excimer. Furthermore, the emission bands of *Anti*-DPyB and *Syn*-DPyB did not show a solvatochromism by the solvent polarity, indicating that the charger transfer dynamics do not occur in *Anti*-DPyB and *Syn*-DPyB. Instead, the time-resolved spectroscopic results for *Anti*-DPyB and *Syn*-DPyB show that the excimers rapidly relax to the $^1(T_1T_1)$ state. Thus, we suggest that the $^1(T_1T_1)$ s of *Anti*-DPyB and *Syn*-DPyBB are formed through the excimer state.

On the other hand, it was also proposed that the intermolecular and intramolecular SF dynamics can rapidly occur with a direct process from the S_1 state to the free state due to strong coupling between the S_1 state and the free triplet state.^{59, 60, 61} For example, Dover et al. suggested that the SF channel is dominated by a direct mechanism from the S_1 state and the formation of the excimer state inhibits efficient SF dynamics.⁶¹ Thus, we also explored the possibility that our data from *Anti*-DPyB and *Syn*-DPyB can be explained using the same direct mechanism by applying the kinetic analysis with the reaction schemes compatible with the direct SF mechanism. Those reaction schemes involving a direct SF process did not accurately reproduce the measured TA spectra for *Anti*-DPyB in *n*-hexane and acetonitrile or yielded an unphysical SADS curve (Supplementary Information and Supplementary Figure S23). This result indicates that the coupling between the S_1 state and the free triplet state in CLDs such as *Anti*-DPyB and *Syn*-DPyB is weaker than in the molecules that showed such direct SF processes, although further systematic studies are needed to confirm this hypothesis.

3) As pointed out from the second comment, the complicated excited-state dynamics in the proposed dimer should not be originated from their electronic states. Also, it seems not convincing to assign excited-state species regarding very similar TA spectra (FC vs Excimer). Authors should resolve this issue for the revised manuscript.

→ As answered in the comment #2, we assigned the observed multiple components to the transitions from one state to another because formations of all excited states were confirmed by the steady-state and time-resolved spectroscopic measurements. The excitation of 345 nm induces the transition from the ground state to a hot singlet excited state, Franck-Condon (FC) state. As explained in our previous manuscript, the excited molecules in the FC state rapidly relax to the singlet excited state (S_1) via the vibrational relaxation (τ_1). After IVR (τ_1), the excited molecules in the S_1 state have various potential fates, including relaxation to other excited states, such as excimer or triplet excited states, and returning to the ground state via fluorescence ($S_1 \rightarrow S_0$). The observation of the excimer fluorescence for *Anti*-DPyB and *Syn*-DPyB leads to the interpretation that a part of the excited molecules in the S_1 state relaxes to the excimer state. In this regard, we assigned the τ_2 to the excimer formation via the $S_1 \rightarrow$ excimer transition. In the case of *Anti*-DPyB, the $S_1 \rightarrow$ excimer transition accompanies the twisting motion between Py and phenyl moiety. On one hand, the long-lived species in *Anti*-DPyB and *Syn*-DPyB are formed with τ_3 time constants. The long-lived species of *Anti*-DPyB and *Syn*-DPyB show structured TA spectra, similar to the T_1 -to- T_n absorption spectra for carbonylpyrenes reported by Rajagopal et al.⁴⁰ Furthermore, as shown in Supplementary Figure S14, the TA signals of *Anti*-DPyB observed at >5 ns resemble the T_1 -to- T_n absorption spectrum of 1-(2-bromophenyl)pyrene measured in dichloromethane (DCM). Therefore, the long-lived species observed in *Anti*-DPyB and *Syn*-DPyB are T_1 (free triplet state) or a similar state that gives an absorption spectra similar to that of T_1 -to- T_n . Consequently, we attribute the τ_3 to the excimer $\rightarrow ^1(T_1T_1)$ transition. Meanwhile, the τ_4 time constant (>10 ns) can be assigned to a $^1(T_1T_1) \rightarrow S_0$ or the $^1(T_1T_1) \rightarrow 2T_1$ transition. *Anti*-DPyB in *n*-hexane and acetonitrile show

a weak and broad absorption band around 445 nm at a few microseconds time delays (Supplementary Figure S12), suggesting the presence of a long-lived species such as a triplet species. In contrast, we could not observe any absorption band for *Syn*-DPyB in both *n*-hexane and acetonitrile, suggesting that there are no free triplets at the microsecond time scales. These results indicate that the dissociation dynamics of $^1(T_1T_1)$'s in *Syn*-DPyB are less favorable than in *Anti*-DPyB. In this regard, the last relaxation time (>10 ns) observed from *Anti*-DPyB is attributed to the dissociation dynamics of $^1(T_1T_1)$, although we could not precisely determine it because of the limited range of investigated delay times. We denote the time constant corresponding to the dissociation dynamics of $^1(T_1T_1)$ as τ_5 .

On the other hand, the reviewer doubted our assignment for the excited-state species because of the similarity of SADS of *Anti*-DPyB and *Syn*-DPyB. However, as shown in the Figure below (Figure R1), SADS curves for the FC and excimer of *Anti*-DPyB are greatly different from each other, indicating that the two SADS curves come from different species. Meanwhile, as mentioned by the reviewer, SADS curves for FC and excimer of *Syn*-DPyB look similar. However, a closer look at two SADS curves for *Syn*-DPyB reveals differences (Figure R2). The small difference in two SADS curves of *Syn*-DPyB is probably due to the rigid structure of *Syn*-DPyB in the ground and excited states and the small energy difference between the states.

Figure R1. Species-associated difference spectra for FC and excimer of *Anti*-DPyB in *n*-hexane (left) and acetonitrile (right).

Figure R2. Species-associated difference spectra for FC and excimer of *Syn*-DPyB in *n*-hexane (left) and acetonitrile (right).

Responses to the comments from Reviewer #3

The authors report the singlet fission dynamics of two different covalently-linked pyrene dimers. The number of SF systems using pyrene derivatives is extremely limited and the research concept may be relatively high. However, the experiment data already described in the text and supporting information are not sufficient to conclude the occurrence of singlet fission and its dynamics, and a significant amount of additional experiments are needed. In addition, a few important findings have been reported on the control of steric configuration for high-yield intramolecular SF and generation of free triplets using covalent-linked dimers, but they have not been fully explained in the introduction section. Authors should reconsider the research content following the reviewer's comments.

→ We appreciate the valuable comments and suggestions on our study. In the revised version, we tried to improve the manuscript by taking into account the comments of the reviewer as detailed in the following.

1. The degree of electronic coupling between two pyrene chromophores (Anti-DPyB vs. Syn-DPyB) should be estimated by absorption spectra and/or electrochemical method. If impossible, DFT calculations can provide some quantitative information.

→ We appreciate the reviewer for pointing this out. According to the reviewer's comment, we evaluated the degree of electronic coupling between the two Py chromophores in *Anti*-DPyB and *Syn*-DPyB using an electrochemical method. As shown in Supplementary Figure S14, the cyclic voltammograms for the reduction of *Anti*-DPyB and *Syn*-DPyB in THF show two separated peaks, whereas the cyclic voltammogram of Py exhibits a single peak. The electrochemical parameters of *Anti*-DPyB and *Syn*-DPyB evaluated by cyclic voltammograms are summarized in Table 3. From the cyclic voltammograms, the splitting energies ($E_{\text{red}2} - E_{\text{red}1}$) of *Anti*-DPyB and *Syn*-DPyB are determined to be 0.09 and 0.12 V, respectively. Compared to *Syn*-DPyB, the lower splitting energy of *Anti*-DPyB indicates that the electronic coupling between the two Py moieties in *Anti*-DPyB is relatively weaker than that in *Syn*-DPyB. This result is consistent with the steady-state spectroscopic results. We added this result and discussion in the revised manuscript. The summarized data were added in Table 3 and Figure S14.

“To further elucidate the interaction between the two Py moieties in *Anti*-DPyB and *Syn*-DPyB, we evaluated the degree of electronic coupling between the two Py chromophores in *Anti*-DPyB and *Syn*-DPyB using an electrochemical method. As shown in Supplementary Figure S14, the cyclic voltammograms for the reduction of *Anti*-DPyB and *Syn*-DPyB in THF show two separated peaks, whereas the cyclic voltammogram of Py exhibits a single peak. The electrochemical parameters of *Anti*-DPyB and *Syn*-DPyB evaluated by cyclic voltammograms are summarized in Table 3. From the cyclic voltammograms, the splitting energies ($E_{\text{red}2} - E_{\text{red}1}$) of *Anti*-DPyB and *Syn*-DPyB are determined to be 0.09 and 0.12 V, respectively. Compared to *Syn*-DPyB, the lower splitting energy of *Anti*-DPyB indicates that the electronic coupling between the two Py moieties in *Anti*-DPyB is relatively weaker than that in *Syn*-DPyB. This result is consistent with the steady-state spectroscopic results.”

Table 3. Electrochemical parameters of Py, *Anti*-DPyB and *Syn*-DPyB evaluated by cyclic voltammograms ($E_{\text{ox}1}$ and $E_{\text{ox}2}$: first and second oxidation potentials, $E_{\text{red}1}$ and $E_{\text{red}2}$: first and second reduction potentials)

	$E_{\text{ox1}}^{\text{a,b}}$ (V)	$E_{\text{ox2}}^{\text{a,b}}$ (V)	$E_{\text{red1}}^{\text{a,c}}$ (V)	$E_{\text{red2}}^{\text{a,c}}$ (V)	$E_{\text{ox2}} - E_{\text{ox1}}$ (V)	$E_{\text{red2}} - E_{\text{red1}}$ (V)
Py	0.85		-2.22			
Anti -DPyB	0.83	0.98	-2.07	-2.16	0.15	0.09
Syn -DPyB	0.85	1.12	-2.06	-2.18	0.27	0.12

^a Determined by cyclic voltammetry (vs SCE)

^b Measured in CH_2Cl_2

^c Measured in THF

Figure S14. Cyclic voltammograms of Py, *Anti*-DPyB, and *Syn*-DPyB. (A) Cyclic voltammograms for the oxidation wave of Py, *Anti*-DPyB, and *Syn*-DPyB in CH_2Cl_2 with 0.1 M $n\text{Bu}_4\text{NPF}_6$ as the supporting

electrolyte. Scan rate: $50 \text{ mV}\cdot\text{s}^{-1}$. (B) Cyclic voltammograms for the reduction wave of Py, *Anti*-DPyB, and *Syn*-DPyB in THF with $0.1 \text{ M } n\text{Bu}_4\text{NPF}_6$ as the supporting electrolyte. Scan rate: $50 \text{ mV}\cdot\text{s}^{-1}$.

2. Authors should measure phosphorescence spectra of these dimers or a reference monomer to experimentally demonstrate the triplet energies of pyrene derivatives in this study. Do these molecules satisfy the energy level matching condition between singlet and triplet excited states?

→ We appreciate the reviewer's suggestion. In our previous manuscript, we proposed that the SF dynamics in *Anti*-DPyB and *Syn*-DPyB are endothermic reactions (Figure 6) based on the reported singlet energy (E_{S1}) and triplet energy (E_{T1}) of Py, which are 3.33 and 2.0 eV, respectively. Considering the reviewer's comment, we measured emission spectra of Py, Ph-Py, *Anti*-DPyB, and *Syn*-DPyB in MTHF containing iodomethane at 77 K to accurately determine the E_{S1} and E_{T1} values of Py, Ph-Py, *Anti*-DPyB and *Syn*-DPyB. As shown in the figure below (Figure S8), all four compounds show dual emission bands at around 370–550 and 580–800 nm, corresponding to fluorescence and phosphorescence, respectively. The E_{S1} values of Py, Ph-Py, *Anti*-DPyB, and *Syn*-DPyB are determined to be 3.3, 3.3, 3.3, and 3.1 eV, respectively. From the phosphorescence spectra, the E_{T1} values of Py, Ph-Py, *Anti*-DPyB, and *Syn*-DPyB are determined to be 2.10, 2.03, 2.04, and 1.87 eV, respectively. The emission spectra of Py, Ph-Py, *Anti*-DPyB, and *Syn*-DPyB measured at 77 K reveal that (i) the E_{T1} values of *Anti*-DPyB and *Syn*-DPyB are lower than that of Py and (ii) $2E_{T1}$ values of *Anti*-DPyB (4.08 eV) and *Syn*-DPyB (3.74 eV) are higher than their E_{S1} values (3.3 and 3.1 eV, respectively), suggesting that the SF processes in *Anti*-DPyB, and *Syn*-DPyB are endothermic reactions (Figure 6). We added this result and discussion in the revised manuscript as follows.

“To accurately determine the singlet energy (E_{S1}) and triplet energy (E_{T1}) values of Py, Ph-Py, *Anti*-DPyB, and *Syn*-DPyB, we also measured emission spectra of Py, Ph-Py, *Anti*-DPyB, and *Syn*-DPyB in MTHF containing iodomethane at 77 K. As shown in Supplementary Figure S8, all four compounds show dual emission bands at around 370–550 and 580–800 nm, corresponding to fluorescence and phosphorescence, respectively. The E_{S1} values of Py, Ph-Py, *Anti*-DPyB, and *Syn*-DPyB are determined to be 3.3, 3.3, 3.3, and 3.1 eV, respectively. From the phosphorescence spectra, the E_{T1} values of Py, Ph-Py, *Anti*-DPyB, and *Syn*-DPyB are determined to be 2.10, 2.03, 2.04, and 1.87 eV, respectively.”

“As shown in Supplementary Figure S8, the emission spectra of Py, Ph-Py, *Anti*-DPyB, and *Syn*-DPyB measured at 77 K reveal that (i) the E_{T1} values of *Anti*-DPyB and *Syn*-DPyB are lower than that of Py and (ii) $2E_{T1}$ values of *Anti*-DPyB (4.08 eV) and *Syn*-DPyB (3.74 eV) are higher than their E_{S1} values (3.3 and 3.1 eV, respectively), suggesting that the SF processes in *Anti*-DPyB, and *Syn*-DPyB are endothermic reactions.”

Figure S8. Emission spectra of Py, Ph-Py, *Anti*-DPyB, and *Syn*-DPyB measured in MTHF containing iodomethane at 77 K.

3. Although the authors propose a kinetic model as shown in Figure 6, the shape of the spectrum in each state is very similar, and the reliability of the relation to the proposed kinetic model is not sufficient.

→ We observed five and four excited states for *Anti*-DPyB and *Syn*-DPyB using the steady-state and time-resolved spectroscopic measurements, respectively. Thus, we assigned the observed multi-components to the transition between the states. The excitation of 345 nm induces the transition from the ground state to a hot singlet excited state, the Franck-Condon (FC) state. As explained in our previous manuscript, the excited molecules of the FC state rapidly relax to the singlet excited state (S_1) via the vibrational relaxation (τ_1). After IVR (τ_1), the excited molecules in the S_1 state have various potential fates, including relaxation to other excited states, such as excimer or triplet excited states, and returning to the ground state via fluorescence ($S_1 \rightarrow S_0$). The observation of the excimer fluorescence for *Anti*-DPyB and *Syn*-DPyB leads to the interpretation that a part of the excited molecules in the S_1 state relaxes to the excimer state. In this regard, we assigned the τ_2 to the excimer formation via the $S_1 \rightarrow$ excimer transition. In the case of *Anti*-DPyB, the $S_1 \rightarrow$ excimer transition accompanies the twisting motion between Py and phenyl moiety. On one hand, the long-lived species in *Anti*-DPyB and *Syn*-DPyB are formed with τ_3 time constants. The long-lived species of *Anti*-DPyB and *Syn*-DPyB show structured TA spectra, similar to the T_1 -to- T_n absorption spectra for carbonylpyrenes reported by Rajagopal et al. Furthermore, as shown in Supplementary Figure S14, the TA signals of *Anti*-DPyB observed at > 5 ns resemble the T_1 -to- T_n absorption spectrum of 1-(2-bromophenyl)pyrene measured in dichloromethane (DCM). Therefore, the long-lived species observed in *Anti*-DPyB and *Syn*-DPyB are T_1 (free triplet state) or a similar state that gives an absorption spectra similar to that of T_1 -to- T_n . Consequently, we attribute the τ_3 to the excimer $\rightarrow {}^1(T_1T_1)$ transition. Meanwhile, the τ_4 time constant (>10 ns) can be assigned to a ${}^1(T_1T_1) \rightarrow S_0$ or the ${}^1(T_1T_1) \rightarrow 2T_1$ transition. *Anti*-DPyB in *n*-hexane and acetonitrile show a weak and broad absorption band around 445 nm at a few microseconds time delays

(Supplementary Figure S12), suggesting the presence of a long-lived species such as a triplet species. In contrast, we could not observe any absorption band for *Syn*-DPyB in both *n*-hexane and acetonitrile, suggesting that there are no free triplets at the microsecond time scales. These results indicate that the dissociation dynamics of $^1(T_1T_1)$'s in *Syn*-DPyB are less favorable than in *Anti*-DPyB. In this regard, the last relaxation time (>10 ns) observed from *Anti*-DPyB is attributed to the dissociation dynamics of $^1(T_1T_1)$, although we could not precisely determine it because of the limited range of investigated delay times. We denote the time constant corresponding to the dissociation dynamics of $^1(T_1T_1)$ as τ_5 .

On the other hand, the reviewer doubted our assignment for the excited-state species because of the similarity of SADS of *Anti*-DPyB and *Syn*-DPyB. However, as shown in Figure 4a, the five SADS curves for *Anti*-DPyB are greatly different from each other. If you look closely at the SADS curves of *Syn*-DPyB, you can see that there is a difference. The small difference in SADS curves of *Syn*-DPyB is probably due to the rigid structure of *Syn*-DPyB in the ground and excited states and small energy difference between each state. We added the following sentence in the revised manuscript.

“The small differences among SADS curves of *Syn*-DPyB are probably due to the rigid structure of *Syn*-DPyB in the ground and excited states and the small energy differences between the states.

Figure 4. Species-associated difference spectra and population changes of intermediates for *Anti*-DPyB obtained from the kinetic analysis for Kinetic Model (2) in *n*-hexane and acetonitrile. (A, B) Species-associated difference spectra in (A) *n*-hexane and (B) acetonitrile. (C, D) Population changes of intermediates in (C) *n*-hexane and (D) acetonitrile. The solid lines are the concentrations obtained from the kinetics analysis. The open circles represent the measure time delays.

Figure 5. Species-associated difference spectra and population changes of intermediates for *Syn*-DPyB obtained from the kinetic analysis for Kinetic Model (4) in *n*-hexane and acetonitrile. (A, B) Species-associated difference spectra in (A) *n*-hexane and (B) acetonitrile. (C, D) Population changes of intermediates in (C) *n*-hexane and (D) acetonitrile. The solid lines are the concentrations obtained from the kinetics analysis. The open circles represent the measure time delays.

4. Associated with above questions, authors should measure time-resolved EPR and directly assign the TT and $T+T$ states, respectively.

→ We appreciate the reviewer for pointing this out. The time-resolved EPR technique is a powerful tool to assign the ${}^1(T_1T_1)$ and $T+T$ states. However, it is known that the time-resolved EPR technique is difficult to provide information on fast singlet fission due to its low temporal resolution (> 100 ns). From this point of view, we speculate that it is difficult to detect the ${}^1(T_1T_1)$ species, which have a short lifetime (*Anti*-DPyB: tens of nanoseconds, *Syn*-DPyB: a few nanoseconds) as an intermediate formed in the SF process of *Anti*-DPyB and *Syn*-DPyB. On one hand, the long-lived free triplet generated by the dissociation dynamics of ${}^1(T_1T_1)$ can be detected using nanosecond TA spectroscopy. Indeed, *Anti*-DPyB in *n*-hexane and acetonitrile show a weak and broad absorption band around 445 nm at a few microseconds time delays (Supplementary Figure S12), suggesting the presence of a long-lived species such as a triplet species. In contrast, *Syn*-DPyB does not exhibit any absorption band in both *n*-hexane and acetonitrile, indicating there are no long-lived species such as a triplet species at the μ s - ms time scales. The weak and broad absorption band around 445 nm observed from *Anti*-DPyB is almost identical to the T_1 -to- T_n absorption band of *Anti*-DPyB measured in iodomethane (also please see our

response to the comment #5), suggesting that the long-lived species of *Anti-DpyB* observed by ns TA experiment in *n*-hexane and acetonitrile are attributed to free triplets generated by the dissociation dynamics of $^1(T_1T_1)$.

5. Authors should carefully confirm the T-T absorption spectra of Anti-DPyB and Syn-DPyB by energy transfer from energy donor dye to these Pyrene dimers.

→ To confirm the dissociation dynamics of $^1(T_1T_1)$ in the SF process of *Anti-DPyB* and *Syn-DPyB*, we measured nanosecond TA spectra of *Anti-DPyB* and *Syn-DPyB* in *n*-hexane and acetonitrile with 355 nm excitation. *Anti-DPyB* in *n*-hexane and acetonitrile show a weak and broad absorption band around 445 nm at a few microseconds time delays (Supplementary Figure S12), suggesting the presence of a long-lived species such as a triplet species. In contrast, *Syn-DPyB* does not exhibit any absorption band in both *n*-hexane and acetonitrile, indicating that no long-lived species, such as a triplet species, exist at the μs - ms time scale. *Syn-DPyB* does not exhibit any absorption band in both *n*-hexane and acetonitrile, indicating that no long-lived species, such as a triplet species, exist at the μs - ms time scale. As shown in Supplementary Figures S13 and S16, the absorption bands around 445 nm of *Anti-DPyB* are similar to the T_1 -to- T_n absorption spectrum of 1-(2-bromophenyl)pyrene measured in DCM. Thus, in our previous manuscript, we assigned the long-lived species observed by nanosecond TA spectra of *Anti-DPyB* to the free triplet species generated by the dissociation dynamics of $^1(T_1T_1)$. To further confirm that the transient absorption band around 445 nm of *Anti-DPyB* corresponds to the T_1 -to- T_n absorption, we additionally performed the nanosecond TA experiment for *Anti-DPyB* in iodomethane. Generally, measuring the absorption bands for the T_1 -to- T_n absorption of aromatic molecules, such as Py, at room temperature in solution is difficult because of the large radiative rate constant from S_1 and poor ISC from S_1 to T_1 . The presence of heavy atoms, such as Br and I, enhances the triplet yields of aromatic molecules in solution due to the heavy atom effect on the ISC. For this reason, we used iodomethane as a solvent to maximize the heavy atom effect. Indeed, as shown in Supplementary Figure S17A, *Anti-DpyB* in iodomethane exhibits an intense absorption band in the range of 350–600 nm at a 1 μs time delay, which is almost identical to those of *Anti-DPyB* measured in *n*-hexane and acetonitrile. Furthermore, we performed the nanosecond TA experiment for Py, Ph-Py, and *Syn-DpyB* in iodomethane. Their TA spectra show absorption bands in the range of 350–600 nm at a 1 μs time delay (Supplementary Figure S17B), whereas no absorption bands were observed in *n*-hexane and acetonitrile. These results indicate that the triplet yields of Py, Ph-Py, *Anti-DpyB*, and *Syn-DpyB* in iodomethane are increased by the heavy atom, resulting in the observation of their T_1 -to- T_n absorptions. The TA spectra from Ph-Py and *Syn-DPyB* are highly similar. Thus, we suggest that the chemical species of *Anti-DpyB* observed at the μs - ms time scale are the free triplets generated by the dissociation dynamics of $^1(T_1T_1)$. We added this result and discussion in the revised manuscript as follows.

“To confirm our interpretation, we additionally performed the nanosecond TA experiment for *Anti-DPyB* in iodomethane. Generally, measuring the absorption bands for the T_1 -to- T_n absorption of aromatic molecules, such as Py, at room temperature in solution is difficult because of the large radiative rate constant from S_1 and poor ISC from S_1 to T_1 . The presence of heavy atoms, such as Br and I, enhances the triplet yields of aromatic molecules in solution due to the heavy atom effect on the ISC. For this reason, we used iodomethane as a solvent to maximize the heavy atom effect. Indeed, as shown in Supplementary Figure S17A, *Anti-DpyB* in iodomethane exhibits an intense absorption band in the range of 350–600 nm at a 1 μs time delay, which is almost identical to those of *Anti-DPyB* measured in *n*-hexane and acetonitrile. Furthermore, we performed the nanosecond TA experiment for Py, Ph-Py, and *Syn-DpyB* in iodomethane. Their TA spectra show absorption bands in the range of 350–600 nm at a 1 μs time delay (Supplementary Figure S17B), whereas no absorption bands were observed

in *n*-hexane and acetonitrile. These results indicate that the triplet yields of Py, Ph-Py, *Anti*-DpyB, and *Syn*-DpyB in iodomethane are increased by the heavy atom, resulting in the observation of their T₁-to-T_n absorptions. The TA spectra from Ph-Py and *Syn*-DPyB are highly similar. Thus, we suggest that the chemical species of *Anti*-DpyB observed at the μs - ms time scale are the free triplets generated by the dissociation dynamics of ¹(T₁T₁).”

Figure S17. (A) Comparison of nanosecond transient absorption spectra of *Anti*-DPyB in *n*-hexane (top), acetonitrile (middle), and iodomethane (bottom). (B) Nanosecond transient absorption spectra of Py (black), Ph-Py (red), *Anti*-DPyB (blue), and *Syn*-DPyB (pink) in iodomethane.

6. Associated with the above content, the molar absorption coefficients should be estimated. Then, the triplet yield of individual triplet excited states should be calculated.

→ According to the reviewer comment, We estimated the triplet quantum yield (Φ_T) of *Anti*-DPyB in *n*-hexane and acetonitrile using nanosecond TA spectroscopy. Φ_T was calculated using the following equation,

$$\Phi_T = \frac{\Delta A_{\text{Sample}}}{\Delta A_{\text{Ref}}} \frac{\varepsilon_{T_{\text{Ref}}}}{\varepsilon_{T_{\text{Sample}}(\text{Py})}} \frac{Abs_{\text{Ref}}}{Abs_{\text{Sample}}} \cdot \Phi_{T_{\text{Ref}}} \quad (\text{S1})$$

where ΔA_{Sample} and ΔA_{Ref} are the delta absorbances of the sample and the reference measured by nanosecond TA experiment, respectively. Φ_T and $\Phi_{T_{\text{Ref}}}$ represent the triplet quantum yields of the sample and the reference sample, respectively. $\varepsilon_{T_{\text{Sample}}(\text{Py})}$ and $\varepsilon_{T_{\text{Ref}}}$ are the triplet extinction coefficients of pyrene (Py) and the reference sample, respectively. Abs_{Sample} and Abs_{Ref} are the absorbances of the sample and the reference sample at 355 nm, respectively. The triplet quantum yield ($\Phi_{T_{\text{Ref}}} = 1$) of benzophenone was used for $\Phi_{T_{\text{Ref}}}$. It is known that the ε_T value of Py in benzene is 20900 M⁻¹ cm⁻¹ at 420 nm and the ε_T value of benzophenone in benzene is 7630 at 532.5 nm.³ Since it is generally known that the effect of solvents on the triple extinction coefficient of a solute molecule is small, we used the ε_T values of Py and benzophenone measured in benzene to determine the Φ_T values of *Anti*-DPyB in *n*-hexane and acetonitrile. From the nanosecond TA experiment, the Φ_T values of *Anti*-DPyB in *n*-hexane and acetonitrile are determined to be 44.1 and 17.5%, respectively.

“We estimated the triplet quantum yield (Φ_T) of *Anti*-DPyB in *n*-hexane and acetonitrile using the data from nanosecond TA spectroscopy. The Φ_T values of *Anti*-DPyB in *n*-hexane and acetonitrile are determined to be 44.1% and 17.5%, respectively (Supplementary Information). [...]”

A more detailed discussion was added to Supplementary Information as follows.

“Triplet quantum yield (Φ_T) of *Anti*-DpyB: We estimated the triplet quantum yield (Φ_T) of *Anti*-DPyB in *n*-hexane and acetonitrile using nanosecond TA spectroscopy. Φ_T was calculated using the following equation,

$$\Phi_T = \frac{\Delta A_{\text{Sample}}}{\Delta A_{\text{Ref}}} \frac{\varepsilon_{T_{\text{Ref}}}}{\varepsilon_{T_{\text{Sample(Py)}}}} \frac{Abs_{\text{Ref}}}{Abs_{\text{Sample}}} \cdot \Phi_{T_{\text{Ref}}} \quad (\text{S1})$$

where ΔA_{Sample} and ΔA_{Ref} are the delta absorbances of the sample and the reference measured by nanosecond TA experiment, respectively. Φ_T and $\Phi_{T_{\text{Ref}}}$ represent the triplet quantum yields of the sample and the reference sample, respectively. $\varepsilon_{T_{\text{Sample(Py)}}}$ and $\varepsilon_{T_{\text{Ref}}}$ are the triplet extinction coefficients of pyrene (Py) and the reference sample, respectively. Abs_{Sample} and Abs_{Ref} are the absorbances of the sample and the reference sample at 355 nm, respectively. The triplet quantum yield ($\Phi_{T_{\text{Ref}}} = 1$) of benzophenone was used for $\Phi_{T_{\text{Ref}}}$. It is known that the ε_T value of Py in benzene is 20900 M⁻¹ cm⁻¹ at 420 nm and the ε_T value of benzophenone in benzene is 7630 at 532.5 nm.³ Since it is generally known that the effect of solvents on the triple extinction coefficient of a solute molecule is small, we used the ε_T values of Py and benzophenone measured in benzene to determine the Φ_T values of *Anti*-DPyB in *n*-hexane and acetonitrile. From the nanosecond TA experiment, the Φ_T values of *Anti*-DPyB in *n*-hexane and acetonitrile are determined to be 44.1 and 17.5%, respectively.”

7. Page 4 in the text:

Besides, the study on the effect of a molecular configuration, which can affect its excited-state relaxation dynamics, may provide a clue for the optimized spatial arrangements for the high energy conversion efficiency of a real device, but such studies are rare.

The reviewer does NOT think so. Authors should carefully check the recent progress regarding the molecular configuration of covalently-linked dimers.

→ We appreciate the valuable comments and suggestions on our study. According to the reviewer comment, we have thoroughly examined literature that has studied SF dynamics in CLDs. We revised the sentence as follows.

“Despite numerous experimental and theoretical approaches to determining the SF dynamics of CLDs, a full understanding of the parameters influencing the SF dynamics of CLDs is still lacking. Additionally, studies on the effects of a molecular configuration, which can affect its excited-state relaxation dynamics, may provide clues for the optimized spatial arrangements that ensure the high energy conversion efficiency of a real device. In this regard, the role of conformational flexibility in the SF dynamics of CLDs has been studied, but still needs further clarification. Accordingly, in-depth studies are needed to understand their excited-state relaxation dynamics, including SF. From this perspective, [...]”

We also added the following related references in the revised manuscript.

1. Chen M, Bae YJ, Mauck CM, Mandal A, Young RM, Wasielewski MR. Singlet Fission in Covalent Terrylenediimide Dimers: Probing the Nature of the Multiexciton State Using

- Femtosecond Mid-Infrared Spectroscopy. *J. Am. Chem. Soc.* **140**, 9184-9192 (2018). (Ref. #12 in the revised manuscript)
2. Dean JC, *et al.* Photophysical Characterization and Time-Resolved spectroscopy of a Anthradithiophene Dimer: Exploring the Role of Conformation in Singlet Fission. *Phys. Chem. Chem. Phys.* **19**, 23162-23175 (2017). (Ref. #13 in the revised manuscript)
 3. Yamakado T, Takahashi S, Watanabe K, Matsumoto Y, Osuka A, Saito S. Conformational Planarization versus Singlet Fission: Distinct Excited-State Dynamics of Cyclooctatetraene-Fused Acene Dimers. *Angew. Chem. Int. Ed.* **57**, 5438-5443 (2018). (Ref. #14 in the revised manuscript)
 4. Nakamura S, *et al.* Synergetic Role of Conformational Flexibility and Electronic Coupling for Quantitative Intramolecular Singlet Fission. *J. Phys. Chem. C* **125**, 18287-18296 (2021). (Ref. #15 in the revised manuscript)
 5. Paul S, Govind C, Karunakaran V. Planarity and Length of the Bridge Control Rate and Efficiency of Intramolecular Singlet Fission in Pentacene Dimers. *J. Phys. Chem. B* **125**, 231-239 (2021). (Ref. #16 in the revised manuscript)
 6. Lukman S, *et al.* Tuning the Role of Charge-Transfer States in Intramolecular Singlet Exciton Fission through Side-Group Engineering. *Nat. Commun.* **7**, 13622 (2016). (Ref. #37 in the revised manuscript)
 7. Alvertis AM, *et al.* Switching between Coherent and Incoherent Singlet Fission via Solvent-Induced Symmetry Breaking. *J. Am. Chem. Soc.* **141**, 17558-17570 (2019). (Ref. #26 in the revised manuscript)
 8. Dover CB, *et al.* Endothermic Singlet Fission is Hindered by Excimer Formation. *Nat. Chem.* **10**, 305-310 (2018). (Ref. #61 in the revised manuscript)
 9. Pun AB, *et al.* Ultra-Fast Intramolecular Singlet Fission to Persistent Multiexcitons by Molecular Design. *Nat. Chem.* **11**, 821-828 (2019). (Ref. #27 in the revised manuscript)
 10. Fuemmeler EG, *et al.* A Direct Mechanism of Ultrafast Intramolecular Singlet Fission in Pentacene Dimers. *ACS Central Sci.* **2**, 316-324 (2016). (Ref. #59 in the revised manuscript)
 11. Mauck CM, *et al.* Singlet Fission via an Excimer-Like Intermediate in 3,6-Bis(thiophen-2-yl)diketopyrrolopyrrole Derivatives. *J. Am. Chem. Soc.* **138**, 11749-11761 (2016). (Ref. #52 in the revised manuscript)
 12. Fumanal M, Corminboeuf C. Direct, Mediated, and Delayed Intramolecular Singlet Fission Mechanism in Donor-Acceptor Copolymers. *J. Phys. Chem. Lett.* **11**, 9788-9794 (2020). (Ref. #60 in the revised manuscript)
 13. Shizu K, Adachi C, Kaji H. Effect of Vibronic Coupling on Correlated Triplet Pair Formation in the Singlet Fission Process of Linked Tetracene Dimers. *J. Phys. Chem. A* **124**, 3641-3651 (2020) (Ref. #40 in the revised manuscript)

8. Regarding the relationship between excimer state and singlet fission, authors should cite the following paper and discuss it.

“Dover, C. B.; Gallaher, J. K.; Frazer, L.; Tapping, P. C.; Petty Ii, A. J.; Crossley, M. J.; Anthony, J. E.; Kee, T. W.; Schmidt, T. W., *Endothermic singlet fission is hindered by excimer formation. Nat. Chem.* **2018**, *10*, 305-310.”

→ We appreciate the valuable comments and suggestions on our study. We carefully read the paper (Dover et al. *Nat. Chem.* 2018, 10, 305-310) recommended by the reviewer. Dover et al. suggested that the SF channel is dominated by a direct mechanism from the S₁ state and the formation of the excimer state inhibits the efficient SF dynamics. We added the following discussion to the main text.

“The kinetic analyses demonstrate that in *Anti*-DPyB, the $^1(T_1T_1)$ formed through the excimer state slowly dissociates into free triplets with a τ_5 time constant. On the other hand, it was also proposed that the intermolecular and intramolecular SF dynamics can rapidly occur with a direct process from the S_1 state to the free state due to strong coupling between the S_1 state and the free triplet state.^{59, 60, 61} For example, Dover et al. suggested that the SF channel is dominated by a direct mechanism from the S_1 state and the formation of the excimer state inhibits efficient SF dynamics.⁶¹ Thus, we also explored the possibility that our data from *Anti*-DPyB and *Syn*-DPyB can be explained using the same direct mechanism by applying the kinetic analysis with the reaction schemes compatible with the direct SF mechanism. Those reaction schemes involving a direct SF process did not accurately reproduce the measured TA spectra for *Anti*-DPyB in *n*-hexane and acetonitrile or yielded an unphysical SADS curve (Supplementary Information and Supplementary Figure S23). This result indicates that the coupling between the S_1 state and the free triplet state in CLDs such as *Anti*-DPyB and *Syn*-DPyB is weaker than in the molecules that showed such direct SF processes, although further systematic studies are needed to confirm this hypothesis.”

A more detailed discussion was added to Supplementary Information as follows.

“Consideration for direct SF mechanisms from the S_1 state: It was also reported that the intermolecular and intramolecular SF dynamics could rapidly occur with a direct process from S_1 state to the free state due to strong coupling between the S_1 state and the free triplet state.^{4, 5, 6} For example, Dover et al. suggested that the SF channel is dominated by a direct mechanism from the S_1 state and the formation of the excimer state inhibits the efficient SF dynamics.⁶ Thus, we also checked the possibility that our data from *Anti*-DPyB can be explained with the same direct mechanism. Assuming that the assignment of the five intermediates to FC, S_1 , excimer, $^1(T_1T_1)$, and $2T_1$ is valid, we can consider a kinetic model that the direct SF process from the S_1 state to the free triplet state occurs with a time constant of τ_2 . In this kinetic model (Kinetic Model (S1) in Scheme S1), the second time constant (τ_2) observed from *Anti*-DPyB can be assigned to the SF dynamics from the S_1 state to free triplet state while other time constants of τ_1 , τ_3 , and τ_4 are assigned to the IVR, the $S_1 \rightarrow$ excimer transition, and the excimer $\rightarrow ^1(T_1T_1)$ transition, respectively. The $^1(T_1T_1) \rightarrow 2T_1$ transition (τ_5) is excluded to fully reflect the nature of the direct SF process to form the free triplets directly from the S_1 state. We analyzed the TA spectra of *Anti*-DPyB in acetonitrile with this kinetic model (Kinetic Model (S1)). Supplementary Figure S23 shows the SADS curves, population changes for five intermediates (FC, S_1 , excimer, $^1(T_1T_1)$, and $2T_1$), and residuals for *Anti*-DPyB in acetonitrile according to Kinetic Model (S1). The residuals between the experimental TA spectra and the best-fit spectra are not negligible (Supplementary Figure S23), meaning that unlike Kinetic Model (2), Kinetic Model (S1) involving a direct SF process does not reproduce well the measured TA spectra for *Anti*-DPyB in acetonitrile. This result indicates that the direct SF mechanism occurring with the τ_2 time constant does not reproduce well the measured TA spectra for *Anti*-DPyB in acetonitrile. The fit quality could be improved when in this kinetic model (Kinetic Model (S1)), the direct SF process from the S_1 state to the free triplet state is forced to occur with a low reaction yield (50%). But, the SADS for $^1(T_1T_1)$ is strongly negative, which is not possible for ESA from $^1(T_1T_1)$ (data not shown). Thus, this kinetic model can be ruled out. As another possibility, we also considered a kinetic model that the direct SF process and IVR in FC state simultaneously occur with a time constant of τ_1 . In this kinetic model, the first time constant (τ_1) observed from *Anti*-DPyB can be assigned to the direct SF dynamics from the FC state to the free triplet state and IVR while other time constants ($\tau_2 - \tau_4$) are assigned as in the reaction mechanism we propose (Kinetic Model (2)). This direct SF mechanism from the FC state to the free triplet state occurring with the τ_1 time constant does not reproduce well the measured TA spectra for *Anti*-DPyB in acetonitrile (data not shown). These results indicate that the τ_1 and τ_2 time constants observed from *Anti*-DPyB cannot be attributed to the

time constant for the direct SF process. If the SF dynamics in *Anti*-DPyB occurred with a direct mechanism, those CLDs would be likely to show efficient SF dynamics due to the fast SF process from the S_1 state to the free triplet state. However, as explained in the main text, the SF dynamics of *anti*-DPyB shows the low triplet quantum yield in *n*-hexane and acetonitrile (44.1 in *n*-hexane and 17.5% in acetonitrile) and *Syn*-DPyB does not exhibit any absorption band in both *n*-hexane and acetonitrile, suggesting that the SF dynamics in *Anti*-DPyB and *Syn*-DPyB are not efficient. These results suggest that the SF dynamics of *Anti*-DPyB and *Syn*-DPyB cannot be explained by a direct SF mechanism.

The lack of the direct SF process in our data is also evident in the time profiles for transient absorption bands of *Anti*-DPyB around 440 nm, which corresponds to the T-T absorption band showing slow rising features (Figure S23). Fitting the time profiles to an exponential function yields the rising times of > 10 ns and 461 ps in *n*-hexane and acetonitrile, respectively. These rising times indicate that in *Anti*-DPyB, the SF process to form free triplets occurs too slowly to be assigned to the direct SF process. The rising times (> 1 ns) are significantly slower than two time constants (τ_1 and τ_2) assigned τ_1 and τ_2 to the intramolecular vibrational relaxation (IVR) ($\tau_1 = \sim 3$ ps) from the initially populated local excited state (FC state) and the $S_1 \rightarrow$ excimer transition (231 ps in *n*-hexane and 24.3 ps in acetonitrile), respectively. Consequently, our results are more consistent with the scenario that the SF dynamics in *Anti*-DPyB and *Syn*-DPyB proceed through an excimer-mediated process rather than a direct SF mechanism caused by strong coupling between the S_1 state and free triplet state. This result may indicate that the coupling between the S_1 state and free triplet state in CLDs such as *Anti*-DPyB and *Syn*-DPyB is weaker than in the molecules that showed such direct SF processes.”

Scheme S1. Photoinduced reaction schemes for *Anti*-DPyB containing a direct SF process from the S_1 state to the free triplet state. (S_0 : ground state, FC: Franck-Condon state, S_1 : singlet excited state, $^1(T_1T_1)$: correlated triplet pair, and T_1 : free triplet state).

Figure S23. TA spectra analysis for *Anti*-DPyB in acetonitrile with the kinetic model of Kinetic Model (S1). (A) Species-associated difference spectra in acetonitrile. (B) Population changes of intermediates in acetonitrile. The solid lines are the concentrations obtained from the kinetics analysis. The open circles represent the measured time delays. (C) The residual for *Anti*-DPyB in acetonitrile. The substantial residuals indicate that this kinetic model does not explain the experimental data satisfactorily, unlike Kinetic Model (2).

Figure S24. Time profiles for transient absorption bands of *Anti*-DPyB in *n*-hexane (black) and acetonitrile (blue) monitored at 440 nm.

Reviewers' comments:

Reviewer #1 (Remarks to the Author):

I am completely content with the changes made with respect to my comments and have no further remarks.

Reviewer #2 (Remarks to the Author):

The reviewer has carefully examined the response letter. The revised manuscript has been quite improved compared to the previous version. However, the reviewer still has some questions about the author's findings and arguments.

(1) The reviewer wonders why the structural rearrangement process of anti-DPyB is influenced by solvent polarity. Indeed, the excimer formation rate of syn-DPyB correspond to the time constant of ~ 3.0 ps in both solvents.

(2) To confirm the monomer impurity, the authors need to check the fluorescence excitation spectrum.

(3) A comparison of Figures 4 and 5, it is interesting that there is a large difference between TA spectra of Anti-DpyB in n-hexane and ACN, whereas TA spectra of syn-DPyB seem similar irrespective of the solvent polarity. In my opinion, discussion on such discrepancy can give an insight into how SF is mediated by excimer state, unlike Dover's work.

(4) The following statement in the conclusion section, "This result is in contrast to the result predicted from theoretical studies that compared with ortho- and para-linked dimers, meta-linked dimers with a smaller E_b exhibit efficient SF dynamics.", seems to be exaggerated. Indeed, the molecular structures of Anti-DpyB and San-DpyB are highly distorted and the orbital interaction should be significant rather than the effect by the substitution position.

Reviewer #3 (Remarks to the Author):

I have check the revised manuscript and comment by authors. The reviewer strongly suggested the time-resolved EPR measurements because the observation of the quintet state of the correlated triplet is a direct evidence of singlet fission. This molecular system has sufficient lifetime (a few hundred nanoseconds). However, in this stage, there is no DIRECT evidence of singlet fission. Therefore, this manuscript does NOT have sufficient scientific evidence to show the occurrence of singlet fission.

Responses to the comments from Reviewer #1

I am completely content with the changes made with respect to my comments and have no further remarks.

→ We appreciate the positive evaluation of our work.

Responses to the comments from Reviewer #2

The reviewer has carefully examined the response letter. The revised manuscript has been quite improved compared to the previous version. However, the reviewer still has some questions about the author's findings and arguments.

(1) The reviewer wonders why the structural rearrangement process of anti-DPyB is influenced by solvent polarity. Indeed, the excimer formation rate of syn-DPyB corresponds to the time constant of ~ 3.0 ps in both solvents.

→ As mentioned in our previous manuscript, *Anti*-DPyB in the ground state has weak interaction between two Py moieties due to their twisted alignment. Thus, the intramolecular excimer formation in *Anti*-DPyB requires the structural rearrangement of two distant Py moieties. As Py is hydrophobic, two Py moieties in a nonpolar solvent show monomeric behavior, leading to slow excimer formation. Meanwhile, a high polarity solvent facilitates the hydrophobic interaction of two Py moieties, resulting in more efficient and fast excimer formation. Indeed, as shown in Figure 1B, *Anti*-DPyB shows the enhanced emission intensity of the excimer in acetonitrile than in *n*-hexane. Kim et al. showed that in Py-Benz-Py, the twisting motion between Py and phenyl moiety accelerates with the increase of the solvent polarity. As shown in Table 1, the excimer formation of *Anti*-DPyB accompanying the structural rearrangement of two distant Py moieties is faster in acetonitrile than in *n*-hexane. These results suggest that the structural rearrangement process of *Anti*-DPyB for excimer formation is influenced by solvent polarity.

On the other hand, *Syn*-DPyB has a rigid and prestacked configuration even in the ground state. Consequently, the excimer formation in *Syn*-DPyB rapidly occurs without any conformational change. Thus, we suggest that the structural rearrangement process of *Syn*-DPyB for excimer formation is not affected by solvent polarity. Indeed, the emission intensities of *Syn*-DPyB do not show definite solvent dependency, as mentioned by the reviewer.

(2) To confirm the monomer impurity, the authors need to check the fluorescence excitation spectrum.

→ It is known that Py molecule shows a strong fluorescence in solutions. Therefore, if Py molecules are present as impurities in *Anti*-DPyB and *Syn*-DPyB solutions, they may affect the fluorescence properties of *Anti*-DPyB and *Syn*-DPyB. To check the existence of Py impurity in *Anti*-DPyB and *Syn*-DPyB solutions, we measured the fluorescence excitation spectra of *Anti*-DPyB and *Syn*-DPyB in acetonitrile at two peak emission positions (380 and 480 nm). As

can be seen in Figure S8, the fluorescence excitation spectra for the emissions of *Anti*-DPyB and *Syn*-DPyB are significantly different from the absorption spectrum of Py molecule. This result indicates that Py molecules do not exist as impurities in *Anti*-DPyB and *Syn*-DPyB solutions. Although the fluorescence excitation spectra at the two peak emission positions for *Anti*-DPyB are similar (Figure S8A), their maximum peak positions are different; 329 nm for the 380-nm fluorescence excitation spectra and 344 nm for the 480-nm one. Accordingly, the two fluorescence excitation spectra have different ratios of the intensity at 329 nm to that at 344 nm, which suggests that two emission bands of 380 and 480 nm originate from two different emissive states. In addition, the two fluorescence excitation spectra are similar to the absorption spectrum of *Anti*-DPyB. In contrast to *Anti*-DPyB, the fluorescence excitation spectra measured at the two emission wavelengths (380 and 480 nm) of *Syn*-DPyB originating from the two emissive states (monomeric S_1 and excimer states) are significantly different from each other (Figure S8B). The fluorescence excitation spectrum for the 380 nm emission is similar to the absorption spectrum of *Anti*-DPyB, which has a monomeric character, whereas the fluorescence excitation spectrum for the 480 nm emission is similar to the absorption spectrum of *Syn*-DPyB, respectively, indicating that two emission bands of 380 and 480 nm originate from two different emissive states. This result supports our interpretation that the dual emissions (~380 and 480 nm, respectively) from *Anti*-DPyB and *Syn*-DPyB come from two emissive states, Py monomer moieties and the excimer. We added this result and discussion in the revised manuscript as follows.

“We also checked the possibility that Py molecules are present as impurities in *Anti*-DPyB and *Syn*-DPyB solutions. It is known that Py molecule shows a strong fluorescence in solutions. Therefore, if Py molecules are present as impurities in *Anti*-DPyB and *Syn*-DPyB solutions, they may contaminate the fluorescence spectra from the *Anti*-DPyB and *Syn*-DPyB samples. To check this possibility, we measured the fluorescence excitation spectra of *Anti*-DPyB and *Syn*-DPyB in acetonitrile at two emission peak positions (380 and 480 nm). As can be seen in Supplementary Figure S8, the fluorescence excitation spectra from *Anti*-DPyB and *Syn*-DPyB are significantly different from the absorption spectrum of Py molecule. This result indicates that Py molecules do not exist as impurities in *Anti*-DPyB and *Syn*-DPyB solutions. Although the fluorescence excitation spectra at the two peak emission positions for *Anti*-DPyB are similar (Supplementary Figure S8A), their maximum peak positions are different; 329 nm for the 380-nm fluorescence excitation spectra and 344 nm for the 480-nm one. Accordingly, the two fluorescence excitation spectra have different ratios of the intensity at 329 nm to that at 344 nm, which suggests that two emission bands of 380 and 480 nm originate from two different emissive states. In addition, the two fluorescence excitation spectra are similar to the absorption spectrum of *Anti*-DPyB. In contrast to *Anti*-DPyB, the fluorescence excitation spectra measured at the two emission wavelengths (380 and 480 nm) of *Syn*-DPyB originating from the two emissive states (monomeric S_1 and excimer states) are significantly different from each other (Supplementary Figure S8B). The fluorescence excitation spectrum for the 380 nm emission is similar to the absorption spectrum of *Anti*-DPyB, which has a monomeric character, whereas the fluorescence excitation spectrum for the 480 nm emission is similar to the absorption spectrum of *Syn*-DPyB, respectively, indicating that two emission bands of 380 and 480 nm

originate from two different emissive states. This result further supports our conclusion based on the emission spectra that the dual emissions (~380 and 480 nm) from *Anti*-DPyB and *Syn*-DPyB come from two emissive states, Py monomer moieties and the excimer.”

Figure S8. Absorption and fluorescence excitation spectra. A) Absorption and fluorescence excitation spectra of *Anti*-DPyB in acetonitrile. B) Absorption and fluorescence excitation spectra of *Syn*-DPyB in acetonitrile.

(3) A comparison of Figures 4 and 5, it is interesting that there is a large difference between TA spectra of *Anti*-DPyB in *n*-hexane and ACN, whereas TA spectra of *syn*-DPyB seem similar irrespective of the solvent polarity. In my opinion, discussion on such discrepancy can give an insight into how SF is mediated by excimer state, unlike Dover’s work.

→ In fact, this comment is related to the comment #1. Again, as mentioned in our previous manuscript, the excimer formation of *Anti*-DPyB accompanies the structural rearrangement of two distant Py moieties. As the solvent polarity affects the structural rearrangement, the resulting excimers formed in *n*-hexane and acetonitrile will have different configuration due to a different Py-Py interaction. In this regard, *Anti*-DPyB likely has different excited-state structures in *n*-hexane and acetonitrile. The structural difference related to solvent polarity seems to induce a large difference between TA spectra of *Anti*-DPyB in *n*-hexane and

acetonitrile. Meanwhile, *Syn*-DPyB has a rigid and prestacked configuration even in the ground state, and its excimer formation rapidly occurs without any conformational change. Accordingly, the structures of the excited *Syn*-DPyB molecules formed in *n*-hexane and acetonitrile are likely to be similar. Thus, we suggest that the similarity in TA spectra of *Syn*-DPyB in *n*-hexane and acetonitrile is probably due to the similar structures of the excited *Syn*-DPyB molecules formed in *n*-hexane and acetonitrile

(4) The following statement in the conclusion section, “This result is in contrast to the result predicted from theoretical studies that compared with ortho- and para-linked dimers, meta-linked dimers with a smaller E_b exhibit efficient SF dynamics.”, seems to be exaggerated. Indeed, the molecular structures of Anti-DPyB and San-DPyB are highly distorted and the orbital interaction should be significant rather than the effect by the substitution position.

→ We appreciate the valuable comments. Considering the reviewer’s comment, we revised the conclusion as follows.

From: This result is in contrast to the result predicted from theoretical studies that compared with *ortho*- and *para*-linked dimers, *meta*-linked dimers with a smaller E_b exhibit efficient SF dynamics. This finding means that the efficiency of SF dynamics in CLD cannot be predicted only by the substitution position of the chromophore in a CLD. Our results show that compared to *Syn*-DPyB, the efficient SF in *Anti*-DPyB is due to the relatively low electronic coupling owing to the twisted alignment of the two chromophores. This result indicates that the SF dynamics in *ortho*-linked dimers, which generally shows a significant π -orbital overlap between two chromophores, can be modulated by the control of the molecular configuration, consequently suggesting that the molecular geometry of a CLD plays a critical role in its SF dynamics, excimer formation, and ICT.

To: This result differs from the prediction based on theoretical studies proposing that *meta*-linked dimers with a smaller E_b , compared with *ortho*- and *para*-linked dimers, exhibit efficient SF dynamics. This finding suggests that the efficiency of SF dynamics in CLDs cannot be predicted solely by the substitution position of the chromophore in a CLD. As *Anti*-DPyB and *Syn*-DPyB have relatively more distorted structures than previously studied CLDs (Figure 1A), the orbital interaction likely has a much greater effect on their SF dynamics than the substitution position. Indeed, our data show that the relatively more efficient SF in *Anti*-DPyB compared to *Syn*-DPyB is caused by the relatively low electronic coupling between two chromophores owing to their twisted alignment. This result indicates that the SF dynamics in *ortho*-linked dimers, which generally show a significant π -orbital overlap between two chromophores, can be modulated by the control of the molecular configuration, consequently suggesting that the molecular geometry of a CLD plays a critical role in its SF dynamics, excimer formation, and ICT.

Responses to the comments from Reviewer #3

I have checked the revised manuscript and comment by authors. The reviewer strongly suggested the time-resolved EPR measurements because the observation of the quintet state of the correlated triplet is a direct evidence of singlet fission. This molecular system has sufficient lifetime (a few hundred nanoseconds). However, in this stage, there is no DIRECT evidence of singlet fission. Therefore, this manuscript does NOT have sufficient scientific evidence to show the occurrence of singlet fission.

→ To confirm the SF dynamics, we performed time-resolved electron paramagnetic resonance (TR-EPR) measurements for *Anti*-DPyB and *Syn*-DPyB. The TR-EPR technique is a powerful tool for assigning the (T_1T_1) and triplet states. The X-band (9.728 GHz) perpendicular mode TR-EPR spectra of *Anti*-DPyB and *Syn*-DPyB were measured in toluene at 80 K. Figure S20 show EPR spectra of *Anti*-DPyB and *Syn*-DPyB at 128 and 200 ns after photoirradiation. The EPR signals for *Anti*-DPyB and *Syn*-DPyB show the narrow peak splitting of about 34 and 19 mT around 340 mT ($g = 2.002$), respectively. In addition to the narrow peak splitting, *Anti*-DPyB and *Syn*-DPyB exhibit a large peak splitting of about 150 and 115 mT, respectively. The EPR signal for *Anti*-DPyB is well reproduced by the simulation curve for its triplet using EasySpin with zero-field splitting (ZFS) parameters $D = -2399$ MHz and $E = 480$ MHz. Similarly, the EPR signal for *Syn*-DPyB is well reproduced by the simulation curve for its triplet using EasySpin with ZFS parameters $D = -1890$ MHz and $E = 450$ MHz. These consistencies suggest that the EPR signals measured from *Anti*-DPyB and *Syn*-DPyB arise from triplet ($S = 1$) species. To further confirm the origins of TR-EPR signals, the nutation experiment for Q-band (34 GHz) TR-EPR signal of *Anti*-DPyB was measured using the pulse sequence, laser- $T_{\text{delay}}-\pi/2-\tau-\pi-\tau$ -echo, with microwave pulse lengths of 16 - 32 ns and an interpulse time of $\tau = 200$ ns. As shown in Figure S21A, Q-band (34 GHz) TR-EPR spectrum of *Anti*-DPyB shows a microwave E/A polarized pattern similar to X-band EPR spectrum. It is known that in the extremely weak limit of the microwave irradiation field ($B_1 = w_1/g$), the nutation frequency w_n of spin magnetization is simply given by $w_n = w_1[S(S + 1) - m_s(m_s - 1)]^{1/2}$ for a transition $|S, m_s\rangle \leftrightarrow |S, m_s - 1\rangle$. In this experiment, w_1 is 8.5 MHz. The w_n s for EPR peaks of 1131.5, 1195.3, 1229.7, and 1293.0 mT are determined to be all 11.74 MHz. The observed nutation frequency ratios w_n/w_1 are ~ 1.4 , consistent with the theoretical value corresponding to the $T_0 \rightarrow T_{\pm 1}$ transition. This result supports that the EPR signal measured from *Anti*-DPyB is attributed to triplet species. Although we did not perform the nutation measurement on the EPR signal of *Syn*-DPyB, we speculate that the EPR signal measured from *Syn*-DPyB arises from triplet species as well. Unfortunately, we could not observe the EPR signals of (T_1T_1) at a few hundred nanoseconds. We speculate that the absence of EPR signals of (T_1T_1) for *Anti*-DPyB and *Syn*-DPyB at a few hundred nanoseconds is probably due to the shorter lifetimes of (T_1T_1)s than the temporal resolution (~ 120 ns) of our TR-EPR system. The femtosecond TA measurements showed that the time profile for transient absorption bands of *Anti*-DPyB around 440 nm, which well reflects the relaxation kinetics of (T_1T_1), shows slow but distinct rising features (Figure S19A), suggesting that the lifetime of (T_1T_1) for *Anti*-DPyB is longer than 10 ns. Compared with *Anti*-DPyB, *Syn*-DPyB shows a relatively fast decay feature in the time profile for transient absorption bands of 450 nm (Figure S19B). As shown in Table 2, the

(T_1T_1)s of *Syn*-DPyB in *n*-hexane and acetonitrile relax to $2T_1$ and S_0 in parallel with time constants of 6.4 and 4.8 ns, respectively, indicating that the lifetime of (T_1T_1) for *Syn*-DPyB should be significantly shorter than the temporal resolution (~ 120 ns) of our TR-EPR system. Meanwhile, we could not precisely determine the lifetime of (T_1T_1) for *Anti*-DPyB because of the limited range of investigated delay times. Overall, the EPR data lead us to conclude that the lifetime of (T_1T_1) for *Anti*-DPyB should be shorter than the temporal resolution (~ 120 ns) of our TR-EPR system. We modified Figure S19 and added this result and discussion in the revised manuscript as follows.

“To confirm the SF dynamics, we performed time-resolved electron paramagnetic resonance (TR-EPR) measurements for *Anti*-DPyB and *Syn*-DPyB. The X-band (9.728 GHz) perpendicular mode TR-EPR spectra of *Anti*-DPyB and *Syn*-DPyB in toluene were measured at 80 K. Figure S20 show EPR spectra of *Anti*-DPyB and *Syn*-DPyB at 128 and 200 ns after photoirradiation. The EPR signals for *Anti*-DPyB and *Syn*-DPyB show the narrow peak splitting of 34 and 19 mT around 340 mT ($g = 2.002$), respectively. In addition to the narrow peak splitting, *Anti*-DPyB and *Syn*-DPyB exhibit a large peak splitting of 150 and 115 mT, respectively. The EPR signal for *Anti*-DPyB is well reproduced by the simulated curve for its triplet using EasySpin with zero-field splitting (ZFS) parameters of $D = -2399$ MHz and $E = 480$ MHz. Similarly, the EPR signal for *Syn*-DPyB is well reproduced by the simulated curve for its triplet using EasySpin with ZFS parameters of $D = -1890$ MHz and $E = 450$ MHz. These consistencies suggest that the EPR signals measured from *Anti*-DPyB and *Syn*-DPyB arise from triplet ($S = 1$) species. To further confirm the origins of TR-EPR signals, the nutation experiment for Q-band (34 GHz) TR-EPR signal of *Anti*-DPyB was measured using the pulse sequence, laser- $T_{\text{delay}}-\pi/2-\tau-\pi-\tau$ -echo, with microwave pulse lengths of 16 - 32 ns and an interpulse time of $\tau = 200$ ns. As shown in Figure S21A, Q-band (34 GHz) TR-EPR spectrum of *Anti*-DPyB shows a microwave E/A polarized pattern similar to X-band EPR spectrum. It is known that in the extremely weak limit of the microwave irradiation field ($B_1 = w_1/g$), the nutation frequency w_n of spin magnetization is simply given by $w_n = w_1[S(S+1) - m_s(m_s - 1)]^{1/2}$ for a transition $|S, m_s\rangle \leftrightarrow |S, m_s - 1\rangle$. In this experiment, w_1 is 8.5 MHz. The w_n s for EPR peaks of 1131.5, 1195.3, 1229.7, and 1293.0 mT are determined to be all 11.74 MHz. The observed nutation frequency ratios w_n/w_1 are ~ 1.4 , consistent with the theoretical value corresponding to the $T_0 \rightarrow T_{\pm 1}$ transition. This result supports that the EPR signal measured from *Anti*-DPyB is attributed to triplet species. Although the nutation measurement on the EPR signal of *Syn*-DPyB was not performed, we speculate that the X-band EPR signal measured from *Syn*-DPyB arises from triplet species as well. The EPR signals at a few hundred nanoseconds do not show evidence for (T_1T_1). The absence of EPR signals of (T_1T_1) for *Anti*-DPyB and *Syn*-DPyB at a few hundred nanoseconds is probably due to the shorter lifetimes of (T_1T_1)s than the temporal resolution (~ 120 ns) of our TR-EPR system. The femtosecond TA measurements show that the time profile for transient absorption bands of *Anti*-DPyB around 440 nm, which well reflects the relaxation kinetics of (T_1T_1), shows slow but distinct rising features (Figure S19A), suggesting that the lifetime of (T_1T_1) for *Anti*-DPyB is longer than 10 ns. Compared with *Anti*-DPyB, *Syn*-DPyB shows a relatively fast decay feature in the time profile for transient absorption bands of 450 nm (Figure S19B). As shown in Table 2, the (T_1T_1)s of *Syn*-DPyB in

n-hexane and acetonitrile relax to $2T_1$ and S_0 in parallel with time constants of 6.4 and 4.8 ns, respectively, indicating that the lifetime of (T_1T_1) for *Syn*-DPyB should be significantly shorter than the temporal resolution (~ 200 ns) of our TR-EPR system. Meanwhile, we could not precisely determine the lifetime of (T_1T_1) for *Anti*-DPyB because of the limited range of investigated delay times in the TA measurement. Overall, the EPR data lead us to conclude that the lifetime of (T_1T_1) for *Anti*-DPyB should be shorter than the temporal resolution (~ 120 ns) of our TR-EPR system.”

Figure S20. X-band (~ 9.728 GHz) perpendicular mode TR-EPR signals of (A) *Anti*-DPyB and (B) *Syn*-DPyB in toluene at 80 K. A and E indicate absorption and emission, respectively. Simulated spectra of triplets are shown by red lines.

Figure S21. (A) Q-band (34 GHz) TR-EPR signal of *Anti*-DPyB. (B) Fourier transforms of nutations measured at 1131.5, 1195.3, 1229.7, and 1293.0 mT.

Figure S19. Time profiles for transient absorption bands of (A) *Anti*-DPyB and (B) *Syn*-DPyB in *n*-hexane (black) and acetonitrile (blue) monitored at 440 nm and 450 nm, respectively.

In our previous manuscript, we used a kinetic model with four intermediates (FC, S_1 , excimer, and (T_1T_1)) because the nanosecond TA experimental result for *Syn*-DPyB suggested the absence of the free triplet of *Syn*-DPyB at the μs - ms time scale. To the contrary, TR-EPR measurements showed that the end products in photoreactions of *Anti*-DPyB and *Syn*-DPyB are free triplets, suggesting the occurrence of the dissociation of (T_1T_1) 's in *Syn*-DPyB resulting in free triplets with a low yield. Thus, we revised the kinetic model for *Syn*-DPyB. The new kinetic model has five intermediates assigned to FC, S_1 , excimer, (T_1T_1) , and $2T_1$ (see Scheme 1). As shown in Figure R1, the new kinetic model considering the $(T_1T_1) \rightarrow 2T_1$ transition shows smaller residuals between the experimental and the best-fit spectra compared with the previous kinetic model not considering the transition. This result indicates that the dissociation of (T_1T_1) occurs in *Syn*-DPyB. Therefore, we revised Scheme 1 and Figures 5, 6, S25, and S26.

Figure R1. Comparison of TA spectra analyses for *Syn*-DPyB in *n*-hexane and acetonitrile with different kinetic models. (A) TA spectra analysis for *Syn*-DPyB in *n*-hexane. The experimental TA spectra (left), the simulated spectra (middle) by a linear combination of the five SADS curves, and residuals (right). (B) TA spectra analysis for *Syn*-DPyB in acetonitrile. The experimental TA spectra (left), the simulated spectra (middle) by a linear combination of the five SADS curves, and residuals (right).

REVIEWERS' COMMENTS:

Reviewer #3 (Remarks to the Author):

Since the author has performed ESR measurements and the reviewer finally recommend acceptance of the manuscript.

Responses to the comments from Reviewer #3

Since the author has performed ESR measurements and the reviewer finally recommend acceptance of the manuscript.

→ We appreciate the positive evaluation of our work.